# I Can Hear You: Selective Robust Training for Deepfake Audio Detection

**Zirui Zhang**[1], **Wei Hao**[1], **Aroon Sankoh**[2], **William Lin**[3], **Emanuel Mendiola Ortiz**[4],
**Junfeng Yang**[1], **Chengzhi Mao**[5]

[1]Columbia University, [2]Washington University in St. Louis,
[3]New York University, [4]Pennsylvania State University, [5]Rutgers University

{*zz3093, wh2473*}*@columbia.edu, a.j.sankoh@wustl.edu, wl2612@nyu.edu,*
*emo5318@psu.edu, junfeng@cs.columbia.edu, cm1838@rutgers.edu*

## Abstract

Recent advances in AI-generated voices have intensified the challenge of detecting deepfake audio, posing further risks for the spread of scams and disinformation. To tackle this issue, we establish a large scale public voice dataset **DeepFakeVox-HQ**, comprising 1.3 million samples, including 270,000 high-quality deepfake samples from 14 diverse sources. Despite previously reported high accuracy, existing deepfake voice detectors struggle with our diversely collected dataset, and their detection success rates drop even further under realistic corruptions and adversarial attacks. We conduct a holistic investigation into factors that enhance model robustness and show that incorporating a diversified set of voice augmentations is beneficial. Moreover, we find that the best detection models often rely on high-frequency features, which are imperceptible to humans and can be easily manipulated by an attacker. To address this, we propose the **F-SAT**: **F**requency-**S**elective **A**dversarial **T**raining method focusing on high-frequency components. Empirical results demonstrate that our training dataset boosts baseline model performance (without robust training) by 33%, and our robust training further improves accuracy by 7.7% on clean samples and by 29.3% on corrupted and attacked samples, over the state-of-the-art RawNet3 model.

## 1 Introduction

AI-generated voices have become increasingly realistic due to larger datasets and enhanced model capacities (Ju et al., 2024; Neekhara et al., 2024), and they have been used in many important applications (Calahorra-Candao & Martín-de Hoyos, 2024). However, the success of AI-synthesized human voices poses significant security risks, including deepfake voice fraud and scams (Tak et al., 2021; Sun et al., 2023; Yang et al., 2024). A recent CNN report reveals a fraud in Hong Kong where a finance worker sent $25 million to scammers after a video call with a deepfake 'chief financial officer'. The voice was created by an AI model, highlighting the risk of such technology.

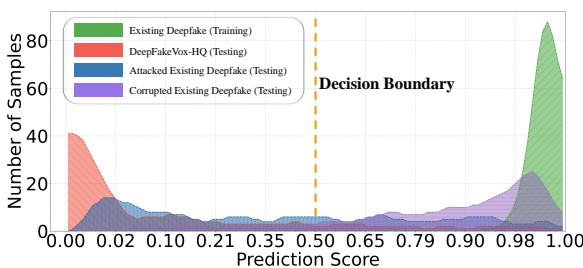

Figure 1: The distribution of deepfake samples over predicted scores using the state-of-the-art detector (Jung et al., 2022) trained on the In-the-Wild dataset (Müller et al., 2022), with a decision boundary at 0.5. Tests on original, corrupted, attacked, and real-world deepfake audio reveal significant shifts in prediction scores, highlighting that training solely on current public datasets without robust training methods leads to poor performance.

Due to the importance of this problem, a number of work has investigated detecting AI-generated audio. Despite previously reported high detection accuracy on public datasets (Todisco et al., 2019;

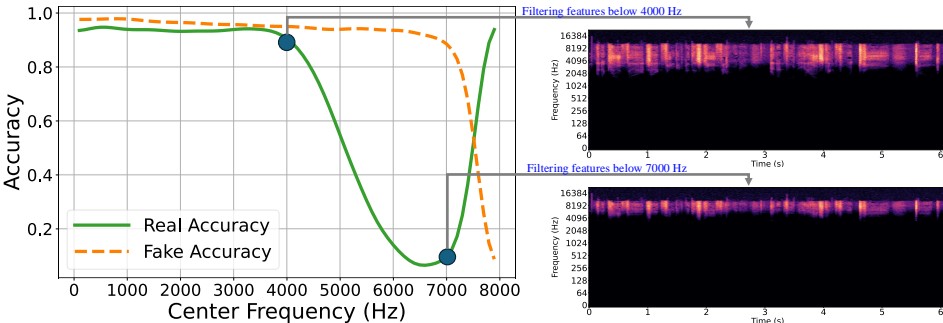

Figure 2: We apply a high-pass filter to audio samples to remove low-frequency components. The x-axis represents the center frequency of the filter applied. Notably, there is a marked decline for RawNet3 Model in detection performance for real audio starting at 4000 Hz and for fake audio at 6000 Hz, suggesting that existing audio models often rely on high frequency signals for prediction.

Frank & Schönherr, 2021), existing deepfake voice detectors perform poorly under real-world conditions (Xu et al., 2020; Müller et al., 2022; Radford et al., 2023). This is because the established benchmarks are often trivial, small, outdated, and homogeneous. Consequently, models trained and validated solely on these datasets fail to generalize to more diverse and challenging real-world deepfake samples. Moreover, deep learning models for audio are particularly vulnerable to adversarial attacks (Szegedy et al., 2013) (Zhang et al., 2019), where an attacker can subtly alter audio inputs in ways that are imperceptible to humans but mislead models into incorrect classifications. Figure 1 illustrates a dramatic shift in the models' prediction scores when exposed to these factors, underscoring the need for more robust training methodologies.

To address the above limitations, we first created the largest deepfake audio dataset to date, *DeepFakeVox-HQ*, including 270,000 high-quality deepfake samples from 14 diverse and distinct origins. We show that simply training on our collected dataset can produce new state-of-the-art models.

Moreover, we find that even the state-of-the-art AI-voice detection model (RawNet3) often depend on high-frequency features to make decisions (see Figure 2), which are imperceptible to humans. On the other hand, the low frequency signals can be heard by humans but are not relied on by the model to make predictions. As a result, natural corruptions in high frequency or attackers can easily manipulate the model by changing the high frequency signals, reducing the detection's robustness.

In an initial study, we observed that standard adversarial training on raw waveforms not only fails to bolster robustness but also diminishes performance on unattacked data. To address these shortcomings, we propose **F**requency-**S**elective **A**dversarial **T**raining (*F-SAT*), which focuses on high-frequency components. Since our adversarial training is targeted, we can mitigate specific vulnerabilities without touching the true features at lower frequencies, thus enhancing the model's resilience to corruptions and attacks while maintaining high accuracy on clean data.

Visualizations and empirical experiments demonstrate that using only our training dataset, we can produce state-of-the-art models, achieving a 33% improvement on the out-of-distribution portion of our test set, which includes 1,000 deepfake samples from the top five AI voice synthesis companies and 600 samples from social media. Additionally, by incorporating random audio augmentations, our model achieves the highest accuracy across 24 different types of corruptions. Furthermore, after applying F-SAT, our model further achieves a 30.4% improvement against adversarial attacks in the frequency domain and an 18.3% improvement against unseen attacks targeting raw waveform data in the time domain.

## 2 RELATED WORK

**AI-synthesized human voice:** AI voice synthesis generally falls into two categories: text-to-speech (TTS) and voice conversion (VC). TTS systems convert written text into spoken audio using a desired voice. This process typically involves three main components: a text analysis module that transforms text into linguistic features, an acoustic model that converts these features into a mel-

| Dataset | #Real Utt | #Fake Utt | Language | Conditions | Year | #Fake Source | Fake Type |
|---------|-----------|-----------|----------|------------|------|--------------|-----------|
| ASVSpoof19 | 12k | 109k | English | Clean | 2019 | 17 | TTS, VC |
| ASVSpoof21 | 22k | 589k | English | Clean, Noisy | 2021 | UNK | TTS, VC |
| WavFake | 14k | 90k | English, Japanese | Clean | 2021 | 7 | TTS |
| ADD2022 | 61k | 251k | Chinese | Clean | 2022 | UNK | TTS, VC |
| In-The-Wild | 20k | 12k | English | Clean, Noisy | 2022 | 19 | TTS |
| LibriSeVoc | 13k | 79k | English | Clean | 2023 | 6 | TTS |
| Our Train | 690k | 640k | English | Clean, Noisy | 2024 | 40 | TTS, VC |
| Our Test | 3k | 3k | English | Clean, Noisy | 2024 | 15 | TTS, VC |

Table 1: Comparison of Deepfake Audio Datasets

spectrogram, and a vocoder. Some of the leading TTS models include StyleTTS (Li et al., 2024), VoiceCraft (Peng et al., 2024), XTTS (Casanova et al., 2024), and Tortoise-TTS (Betker, 2023), and are known for their ability to produce high-quality audio that closely mimics real human speech.

VC models, in contrast, take an audio sample from one person and transform it to sound like another person for the same speech content. Recent VC approaches primarily operate within the mel-spectrum domain (Ju et al., 2024; Shen et al., 2023; Popov et al., 2021), using deep neural networks to shift the mel-spectrograms from the source to the target voice.

**Detection Method:** AI deepfake detection models based on deep learning can be grouped into two main categories: those processing raw audio end-to-end and those analyzing spectrum. The first category includes models like RawNet2, which employs Sinc-Layers (Ravanelli & Bengio, 2018) to extract features directly from waveforms, and RawGAT-ST, which utilizes spectral and temporal sub-graphs (Tak et al., 2021). RawNet3 (Jung et al., 2022), which begins by using parameterized filterbanks (Zeghidour et al., 2018) to extract a time-frequency representation from the raw waveform and then is followed by three backbone blocks with residual connections, a structural approach that sets it apart from ECAPA-TDNN (Desplanques et al., 2020).These models process the audio data in its raw form to capture nuanced details directly impacting model performance.

The second category of AI deepfake detection models involves transforming raw audio into spectrograms for analysis. This process utilizes extracted features such as Mel Frequency Cepstral Coefficients (MFCCs) (Sahidullah & Saha, 2012), Constant Q Cepstral Coefficients (CQCCs) (Todisco et al., 2017), bit-rate (Borzì et al., 2022). The analysis of these features is conducted using traditional machine learning methods like Gaussian Mixture Models (GMMs) (Todisco et al., 2019) or advanced neural networks such as Bidirectional Long Short-Term Memory (Bi-LSTM) (Akdeniz & Becerikli, 2021), ResNet (Alzantot et al., 2019), and Transformers (Zhang et al., 2021). These methods enable deeper and more intricate pattern recognition, enhancing the model's ability to identify and classify deepfake audio accurately.

**Adversarial Attack:** Neural networks are highly vulnerable to nearly imperceptible perturbations, known as adversarial attacks (Szegedy et al., 2013). Although initial studies focused on image models, speech tasks are similarly susceptible. Notably, adversarial attacks generated on spectrograms can deceive 2D CNN models and, when converted back to waveforms, can also effectively fool 1D CNN models (Koerich et al., 2020).

Adversarial training, initially developed for image processing systems (Madry et al., 2017; Mao et al., 2019), has been increasingly adapted to the audio domain (Chen et al., 2023), particularly to enhance the robustness of applications such as Automatic Speech Recognition (ASR) and speaker verification systems. Another defense method, smoothing, based on additive noise masking, has demonstrated great improvements in model robustness for these tasks (Olivier et al., 2021; Cohen et al., 2019). Additionally, a defensive strategy that employs diffusion models to counter adversarial audio attacks (Wu et al., 2023) effectively smooths out perturbations and prevents attackers from altering audio signals without significantly compromising signal quality.

While these techniques enhance robustness, they hurt the detection on unattacked audio, highlighting a trade-off between robustness and accuracy (Zhang et al., 2019; Tsipras et al., 2018). This compromise is particularly critical in scenarios that demand high accuracy and user satisfaction. Furthermore, the generalizability of adversarially trained models to new and unseen attacks remains limited (Rajaratnam et al., 2018), raising questions about their effectiveness in rapidly evolving threat environments.

## 3 DEEPFAKEVOX-HQ

### 3.1 TRAINING DATASET

In this section, we introduce a new training dataset and a rigorous test set. In contrast to prior dataset, our dataset is large, diversified, realistic, and up-to-date, as shown in Table 1. Prior detectors show poor generalization capabilities in realistic settings, as shown in Figure 8. Both our training and testing datasets integrate the latest advancements in AI voice synthesis technologies. Additionally, the testing dataset includes several new models not covered in the training dataset, specifically designed to test the generalization ability of our detection systems.

**High quality deepfake samples:** The limitations of existing public datasets, which often lack high-quality deepfake samples, can potentially impair model performance. To address this, we have investigated more than 30 recent advancements in Text-to-Speech (TTS) and Voice Conversion (VC) models developed in the past few years. Our training dataset now includes deepfake audio samples generated using the top seven TTS models: MetaVoice-1B (Liu et al., 2021), StyleTTS-v2 (Li et al., 2024), VoiceCraft (Peng et al., 2024), WhisperSpeech (Radford et al., 2023), VokanTTS, XTTS-v2 (Casanova et al., 2024), and Elevenlabs. We use four datasets—VCTK (Yamagishi, 2012), LibriSpeech (Panayotov et al., 2015), In-The-Wilds (Müller et al., 2022), and AudioSet (Gemmeke et al., 2017)—to generate deepfake audio. These datasets include both clean, high-quality and noisy, low-quality real audio, ensuring that the deepfake audio produced is highly diverse and accurately reflects real-world conditions.

**Reference data:** For both fake and real audio, having only high-quality samples is insufficient. A broader diversity of samples is essential for the training dataset. Thus, for real audio, we utilize portions from six public audio datasets: VCTK, LibriSpeech, AudioSet, ASRspoof2019 (Todisco et al., 2019), Voxceleb1 (Nagrani et al., 2017), and ASRspoof2021 (Liu et al., 2023), with half consisting of clean audio and the other half of noisy audio. For fake audio, we include two low-quality fake audio datasets: WaveFake (Frank & Schönherr, 2021) and ASRspoof2019 to further enhance the diversity of the training material.

### 3.2 TESTING DATASET

Our test dataset comprises approximately 6,000 samples, with an equal balance between real and fake audio. All data used is legally permissible, as detailed in Section A.1, where we discuss the sources and usage policies in further detail.

Real audio samples are sourced from recent celebrity speeches and conversational videos. For the fake audio, we not only utilize samples created using the seven latest TTS models but have also expanded our dataset to include contributions from eight of the most advanced AI voice synthesis models or commercial software currently available, namely CosyVoice (Du et al., 2024), Resemble, Speechify, LOVO AI, Artlist, and Lipsynthesis. Additionally, this set includes fake audio directly collected from social media platforms like YouTube and X, further enriching the dataset with a diverse range of real-world scenarios. This comprehensive composition is strategically designed to rigorously test the generalization capabilities of our model.

**Insight:** Figure 8 shows that training with previous public datasets yields lower accuracy on ours. Additionally, removing high-quality deepfake samples from our training set significantly also reduces accuracy, highlighting their importance."

## 4 METHOD

In this section, we present our selective adversarial training approach, F-SAT. We also present a taxonomy of the most common corruptions and attacks in audio processing, which can be used for robust evaluation in realistic settings. Additionally, we discuss the implementation of Rand-Augmentation for audio to further enhance the robustness of our detection system.

Figure 3: F-SAT Pipeline

## 4.1 F-SAT: FREQUENCY-SELECTIVE ADVERSARIAL TRAINING

Let $\mathbf{x}$ be a waveform input audio, and $\mathbf{y}$ be its ground-truth category label, To perform classification, neural networks commonly learn to predict the category $\hat{\mathbf{y}} = \mathcal{F}_\theta(\mathbf{x})$ by optimizing the cross-entropy $H(\hat{\mathbf{y}}, \mathbf{y})$ between the predictions and the ground truth. We use RawNet3 (Jung et al., 2022) that can process waveform audio input. The network parameters $\theta$ are estimated by minimizing the expected value of the objective:

$$\mathcal{L}_c(\mathbf{x}, \mathbf{y}) = H(F_\theta(\mathbf{x}), \mathbf{y}),$$

**Time domain Attack:** Adversarial attacks on audio can be directly added to the waveform. Let the additive perturbations be $\boldsymbol{\delta}$. For attacks in the time domain, we directly add $\boldsymbol{\delta}$ to the waveform $\mathbf{x}$. Due to the huge amount of freedom of the attack vector $\boldsymbol{\delta}$, the added attack vectors are often high frequency. We formulate this attack as: $\mathbf{x}' = \mathbf{x} + \boldsymbol{\delta}$.

**Frequency Domain Attack:** Attacks can also be applied in the frequency domain, accessed through a reversible Fourier transformation of the waveform. We transform the waveform $\mathbf{x}$ to the frequency domain $\mathbf{X}$ using the Short-Time Fourier Transform (STFT). The attack modifies $\mathbf{X}$ by adding $\boldsymbol{\delta}$, yielding $\mathbf{X}' = \mathbf{X} + \boldsymbol{\delta}$. The Inverse Short-Time Fourier Transform (ISTFT) then reverts $\mathbf{X}'$ back to the time domain, creating the manipulated audio waveform $\mathbf{x}'$.

**Frequency-Selective Attack:** Compared to time-domain attacks, frequency-domain attacks provide an inductive bias that enables the crafting of more controlled perturbations, such as those within a restricted bandwidth. Building on this, we introduce a Frequency-Selective Attack that targets specific frequency ranges within the magnitude component of a waveform. We define this attack as

$$\mathbf{x}' = s(\mathbf{x}, \boldsymbol{\delta}, f_l, f_u)$$

where $f_l$ and $f_u$ represent the lower and upper frequency boundaries of the range where the attack is to be applied. The procedure can be described as follows:

Starting with STFT result $\mathbf{X}$, we first compute its magnitude component $\mathbf{X}_\rho$ and phase component $\mathbf{X}_\phi$. To map the frequency range to the index range of the STFT spectrum, we calculate the lower and upper boundary indices $r_l$ and $r_u$ of the FFT spectrum using formulas:

$$r_l = \left\lfloor \frac{f_l \times n_{\text{fft}}}{sr} \right\rfloor, \quad r_u = \left\lceil \frac{f_u \times n_{\text{fft}}}{sr} \right\rceil,$$

where $n_{\text{fft}}$ is the number of FFT points, and $sr$ is the sampling rate of the original signal $\mathbf{x}$. We then define a mask function $M$ using a diagonal matrix $\mathbf{D}$ such that

$$\mathbf{D}_{kk} = \begin{cases} 1 & \text{if } r_l \leq k \leq r_u \\ 0 & \text{otherwise} \end{cases}$$

This matrix $\mathbf{D}$ selectively targets the desired frequency components within the specified range $[r_l, r_u]$ for perturbation. The selective perturbation is then defined as:

$$\boldsymbol{\delta_s} = M(\boldsymbol{\delta}, r_l, r_u) = \mathbf{D} \cdot \boldsymbol{\delta}.$$

By adding this perturbation $\boldsymbol{\delta_s}$ to the magnitude component of the spectrogram, the perturbed spectrogram can be represented as:

$$\mathbf{X}' = (\mathbf{X}_\rho + \boldsymbol{\delta_s}) \cdot e^{j\mathbf{X}_\phi} = (\mathbf{X}_\rho + \mathbf{D} \cdot \boldsymbol{\delta}) \cdot e^{j\mathbf{X}_\phi}$$

Finally, we employ the ISTFT to convert the attacked spectrogram $\mathbf{X}'$ back into the time-domain signal $\mathbf{x}'$, thus generating the attacked audio waveform $\mathbf{x}'$.

Moreover, attackers design these worst-case perturbations to disrupt a trained network $\mathcal{F}_\theta$, aiming to maximize misclassifications by optimizing the objective:

$$\mathbf{x}' = \arg\max_{\mathbf{x}'} \mathcal{L}_c(\mathbf{x}', \mathbf{y}), \quad \text{s.t.} \quad \|\mathbf{x}' - \mathbf{x}\|_q < \epsilon,$$

where the perturbation $\boldsymbol{\delta} = \mathbf{x}' - \mathbf{x}$ is constrained by the $q$ norm to be less than $\epsilon$, ensuring minimal deviation.

**F-SAT:** Based on the Frequency-Selective Attack, we propose F-SAT, an adversarial training method which optimizes perturbations in the frequency domain by targeting the magnitude component within specific frequency ranges.

$$\mathbf{x}'_{n+1} = \Pi_{\mathbf{x}+S} \left( x'_n + \alpha \cdot M\left( \text{sgn}\big(\nabla_{\mathbf{x}'}\mathcal{L}(s(\mathbf{x}, \boldsymbol{\delta}, f_l, f_u), y)\big), f_u, f_l \right) \right),$$

where $\Pi_{\mathbf{x}+S}$ represents the projection onto the allowable perturbation set $S$, defined by the condition $\|\mathbf{x}' - \mathbf{x}\|_p \leq \epsilon$. The parameter $\alpha$ denotes the step size of the update, and $\nabla_{\mathbf{x}'}\mathcal{L}$ is the gradient of the loss function with respect to the perturbed input $\mathbf{x}'$.

As illustrated in Figure 3, after $K$ iterations (where $n \geq K$), we feed the most perturbed sample back into the detection model and calculate the cross-entropy loss between it and the ground truth label:

$$\mathcal{L}_{robust}(\mathbf{x}', \mathbf{y}) = H(F_\theta(\mathbf{x}'), \mathbf{y}),$$

Additionally, to maintain a balance between accuracy on original and attacked samples, we train the model using both the original and perturbed samples. We introduce a parameter $\boldsymbol{\gamma}$ to control the trade-off between clean loss and robust loss. The clean loss is defined as:

$$\mathcal{L}_{clean}(\mathbf{x}, \mathbf{y}) = H(F_\theta(\mathbf{x}), \mathbf{y}),$$

The total loss $\mathcal{L}_{total}$ is then computed as a weighted sum of clean and robust losses.

$$\mathcal{L}_{total} = \mathcal{L}_{clean} + \boldsymbol{\gamma} \cdot \mathcal{L}_{robust}$$

Since time domain attacks are often high frequency, by focusing the adversarial training on the high frequency, our approach not only enhances model robustness against frequency-targeted attacks, but also improves defenses against time-domain attacks.

### 4.2 RANDAUGMENT FOR AUDIO

**Realistic Corruptions for Deepfake Audio Generation:** As shown in Figure 9, we present a taxonomy for analyzing the robustness of deepfake audio detection systems that incorporate the most common corruptions and attacks. Corruptions in audio result from unintentional modifications during recording, processing, or transmission, impacting noise levels and frequency responses. Adversarial attacks, in contrast, are intentional manipulations aimed at deceiving detection systems through subtle changes across various attack methodologies.

**RandAugment:** Experiments show that the best detection model, RawNet3, experiences a great drop in accuracy when faced with audio corruptions. Inspired by the image-based RandAugment (Cubuk et al., 2020), which improves model robustness, we adapted this method for audio. We selectively apply $\mathcal{N}$ transformations from the available options, each assigned a uniform probability. An additional probability $p$ determines whether each transformation is applied at a given instance, to balance the accuracy between original and corrupted samples. The magnitude of each transformation is controlled within predefined boundaries, with the intensity randomly sampled from this range.

## 5 EXPERIMENTS

We first compare our approach to existing state-of-the-art methods across three benchmarks and demonstrate improved accuracy. We then assess its robustness against corruption and adversarial attacks. We finally conduct ablation study on enhancements to the detection system's robustness.

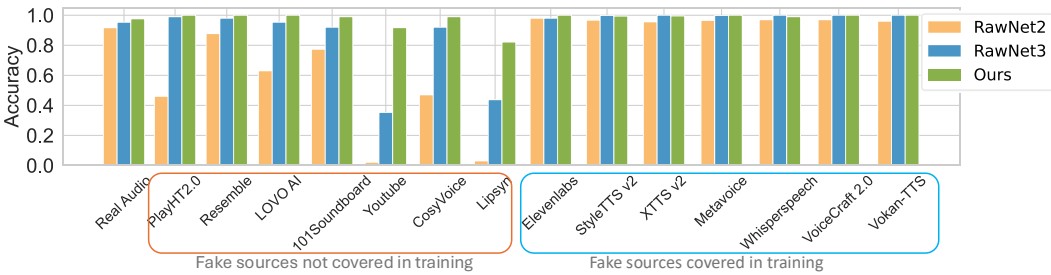

Figure 4: Detailed results across different AI-voice synthesis models on our test set. Fake audio is categorized into sources included and not included in training. Models evaluated include RawNet2, RawNet3, and Ours. Our method outperforms all others across all different sources.

## 5.1 DATASET

**DeepFakeVox-HQ (test)** Our custom test dataset incorporates 14 of the latest and highest quality TTS and VC models to generate fake audio. It also includes fake audio samples directly collected from social media platforms such as YouTube and X, providing a diverse set of real-world scenarios. For the experiments described in this paper, we specifically utilize seven fake sources not present in the training set. However, we also include samples from seven other fake sources used during training to facilitate future research.

**ASVspoof2019** (Todisco et al., 2019) derived from the VCTK base corpus, which includes speech data captured from 107 speakers. It contains three major forms of spoofing attacks, namely synthetic, converted, and replayed speech.

## 5.2 BASELINE

**RawNet2** (Tak et al., 2021) employs Sinc-Layers (Ravanelli & Bengio, 2018) to directly extract features from audio waveforms. These layers function as band-pass filters that enhance the detection of spoofed audio content.

**RawGAT-ST** (Tak et al., 2021) utilizes spectral and temporal sub-graphs integrated with a graph pooling strategy, effectively processing complex auditory environments.

**TE-ResNet** (Zhang et al., 2021) processes synthetic speech detection by first extracting MFCC from input speech. These coefficients are used as features for a CNN that extracts spatial features, followed by a Transformer that analyzes these to detect characteristics of synthetic speech effectively.

**RawNet3** (Jung et al., 2022) begins by using parameterized filterbanks to extract a time-frequency representation from the raw waveform. This is followed by three backbone blocks with residual connections, a structural approach that deviates from ECAPA-TDNN (Desplanques et al., 2020).

## 5.3 MAIN RESULT

**Results on Original Data:** We trained all models on DeepFakeVox-HQ and tested them across two benchmarks: our test set and ASVspoof2019, as shown in Table 2. Our method achieved state-of-the-art results across both benchmarks. Compared with RawNet3, our method shows improvements of 7.8% F1 score on DeepFakeVox-HQ (test), 1.1% F1 score on ASVspoof2019.

Among all baseline models, Rawnet3 performs the best. Therefore, for subsequent evaluations involving corruption and attacks, we will apply robust training methods to Rawnet3 and conduct comparisons.

Figure 4 outlines results across various AI-voice synthesis models on our test set. For the fake sources are included in the training, all models demonstrated high accuracy . However, for unforeseen source, our method significantly outperformed others, particularly those from real-world social media platforms like YouTube and Lipsync, by up to 50% points, highlighting its superior generalization capabilities on out-of-distribution data.

**Result on Corrupted data:** Figure 5a shows detailed results for various corruptions. Our methods are depicted in the green region with F-SAT, and in the red region without F-SAT. They outperform all other baseline methods across 24 types of corruptions, with an average absolute increase of 15.3%

| Approach | DeepFakeVox-HQ | | | ASVspoof2019 | | |
|---|---|---|---|---|---|---|
| | Real Acc | Fake Acc | F1 | Real Acc | Fake Acc | F1 |
| TE-ResNet | 0.859 | 0.433 | 0.629 | 0.894 | 0.973 | 0.933 |
| RawGAT-ST | 0.926 | 0.698 | 0.810 | 0.943 | 0.951 | 0.947 |
| RawNet2 | 0.946 | 0.578 | 0.754 | 0.930 | **0.985** | 0.957 |
| RawNet3 | 0.961 | 0.759 | 0.861 | **0.970** | 0.908 | 0.939 |
| Ours | **0.975** | **0.964** | **0.974** | 0.950 | 0.982 | **0.966** |

Table 2: Comparative performance of our method and baseline models on DeepFakeVox-HQ and ASVspoof2019 test sets, our method outperforms prior approaches. (We ensured our deepfake test set does not overlap with the training set.)

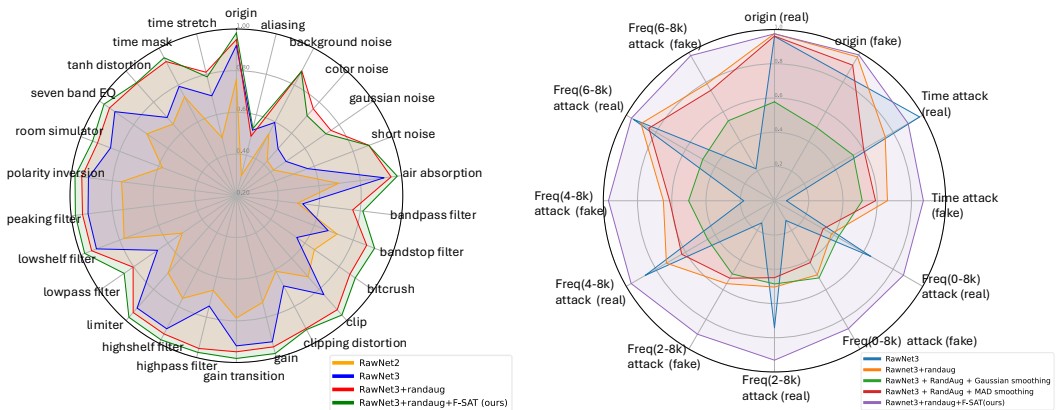

(a) Performance across various corruptions     (b) Performance across different types of attacks

Figure 5: Comparative results showcasing the accuracy of our method against various types of corruptions and attacks. Our method outperforms the baseline across nearly all tested conditions.

points. However, as we can observe, aliasing has the most negative impact on accuracy. During aliasing, audio is intentionally downsampled and then resampled back, which not only removes high-frequency features but also introduces distortions characteristic of aliasing. In addition, low-pass filters, time stretching, and noise also significantly impact performance more than others.

## 5.4 RESULTS ON ATTACKED DATA

To assess our model's robustness against adversarial attacks, we employ the Projected Gradient Descent (PGD) attack, a well-established method in adversarial training. We compare our F-SAT with conventional adversarial training and smoothing defense methods.

**Attack Settings for Evaluation and Perceptibility Assessment:** In the main paper, we evaluate the $l_\infty$-PGD attack on the time domain ($\epsilon = 1 \times 10^{-4}$, $\alpha = 4 \times 10^{-5}$, iter = 2) and on the magnitude component of the frequency domain ($\epsilon = 1 \times 10^{-3}$, $\alpha = 4 \times 10^{-4}$, iter = 2). For the frequency domain, we progressively expand the attacked frequency range to include 6-8 kHz, 4-8 kHz, 2-8 kHz, and 0-8 kHz. The average attack result is calculated across these ranges. More comprehensive attack analyses are provided in Appendix A2. To ensure the imperceptibility of the attacks in both domains, we evaluate the Signal-to-Noise Ratio (SNR) of the attacks compared to the original audio. The SNR values are 58.4 dB for the time domain attack and 68.7 dB for the frequency domain attack, indicating that the attacks remain imperceptible.

**Compared with other defense method:** We compare our Method F-SAT with other defense methods—MAD smoothing (Olivier et al., 2021), Gaussian smoothing (Cohen et al., 2019), and standard adversarial training (Mkadry et al., 2017)—on the RawNet3 model against various adversarial attacks in both time and frequency domains. As shown in Figure 5b, F-SAT outperforms all other methods.

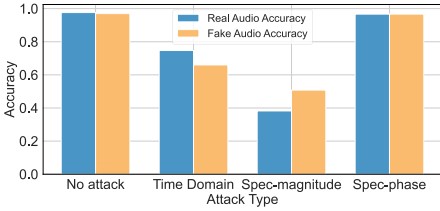 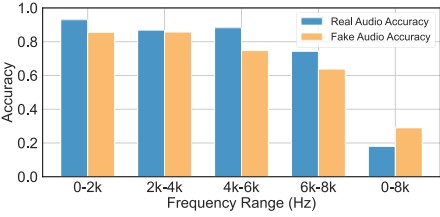

(a) Impact of Different Attack Types      (b) Frequency Range for Attack ($\epsilon = 0.01$)

Figure 6: Exploring Model Vulnerabilities: (a) shows accuracy under under various attack types; (b) details how attacks targeting magnitude in various frequency domains affect accuracy. We show attacks based on magnitude in the frequency domain are highly effective at degrading model performance, with those targeting higher frequency ranges having a greater impact.

## 5.5 ABLATION STUDY AND ANALYSIS

**Choice of attack type for F-SAT:** As discussed in Section 4.2, there are three types of adversarial attacks on audio: time domain, frequency domain on magnitude, and frequency domain on phase. All of these can be employed for adversarial training. However, our focus on targeting the magnitude component in attacks is explained in Figure 6a. Through our evaluation, we found that attacks based on magnitude in the frequency domain are particularly effective at degrading model performance. This is due to the fact that changes in magnitude directly affect the amplitude of audio signals, crucial for maintaining core acoustic features. Such changes alter the sound intensity across frequencies, complicating the model's task of distinguishing key characteristics between real and fake audio. Conversely, phase attacks have minimal impact on the model's predictions, as phase alterations primarily influence spatial audio perception and do not significantly affect feature detection.

We further validate our conclusions experimentally, as shown in Table 3. Our findings indicate that F-SAT excels in adversarial training against attacks on both time and frequency domains. Surprisingly, incorporating adversarial training focused on the time domain appears to diminish overall accuracy on original data and does not effectively enhance robustness against attacks in either domain. Direct attacks in the time domain may disrupt most features critical for differentiating between real and fake audio, potentially overburdening the model during training.

Additionally, the introduction of RandAug to the RawNet3 baseline model significantly enhances accuracy on corrupted data, suggesting that the model benefits from exposure to varied conditions during training.

| Approach | Origin | | | Corruption | | | Attack(Time) | | | Attack(Frequency) | | |
|---|---|---|---|---|---|---|---|---|---|---|---|---|
| | Real Acc | Fake Acc | F1 | Real Acc | Fake Acc | F1 | Real Acc | Fake Acc | F1 | Real Acc | Fake Acc | F1 |
| RawNet3 | 0.961 | 0.759 | 0.861 | 0.947 | 0.556 | 0.720 | **0.979** | 0.035 | 0.134 | 0.804 | 0.083 | 0.202 |
| +RandAug | **0.976** | 0.947 | 0.968 | 0.899 | 0.826 | 0.860 | 0.747 | 0.661 | 0.693 | 0.630 | 0.573 | 0.613 |
| +RandAug+AT(Time) | 0.947 | 0.825 | 0.891 | 0.881 | 0.772 | 0.818 | 0.871 | 0.124 | 0.211 | 0.665 | 0.152 | 0.207 |
| +RandAug+ F-SAT | 0.975 | **0.964** | **0.974** | **0.948** | **0.830** | **0.885** | 0.902 | **0.880** | **0.910** | **0.933** | **0.915** | **0.924** |

Table 3: Ablation study evaluating the impact of our RandAug and F-SAT, comparing their effects against time-domain adversarial training (AT) and phase-based adversarial training. RandAug enhances robustness to corruptions, while F-SAT improves robustness to adversarial attacks.

**Impact of frequency range for F-SAT:** The importance of high-frequency components is underscored in Figure 6b, which shows the model's performance under gradient-based adversarial attacks across different frequency ranges. Attacks targeting higher frequencies notably degrade deepfake detection more than lower frequencies, underscoring the vulnerability of high-frequency features. By focusing attacks on high-frequency regions, we preserve low-frequency integrity, simplifying the model's task of distinguishing deepfake audio features. This approach maintains high accuracy on unattacked data and improves adversarial robustness.

Further analysis on selecting the optimal frequency range for F-SAT is presented in Table 4. Adversarial training within the 4k to 8k frequency range yields the best performance. We observe that increasing the attack frequency range decreases accuracy on original data due to the distortion of more critical features, adding complexity to the model's learning process.

Moreover, we observe that narrowing the frequency range of F-SAT increases the accuracy for attacked fake data while decreasing it for attacked real data. This supports the findings shown in

| Approach | Original | | | Attack (Time) | | | Attack (Frequency) | | |
|---|---|---|---|---|---|---|---|---|---|
| | Real Acc | Fake Acc | F1 | Real Acc | Fake Acc | F1 | Real Acc | Fake Acc | F1 |
| F-SAT (0-8kHz) | 0.977 | 0.818 | 0.898 | **0.978** | 0.795 | 0.877 | 0.970 | 0.751 | 0.862 |
| F-SAT (2-8kHz) | **0.983** | 0.896 | 0.940 | 0.976 | 0.736 | 0.831 | **0.979** | 0.754 | 0.836 |
| F-SAT (4-8kHz) | 0.975 | **0.964** | **0.974** | 0.902 | **0.880** | **0.910** | 0.933 | **0.915** | **0.924** |
| F-SAT (6-8kHz) | 0.971 | 0.974 | 0.952 | 0.912 | 0.850 | 0.879 | 0.923 | 0.896 | 0.907 |

Table 4: The results of frequency range selections for F-SAT. 4-8k is the most effective range.

Figure 2, which highlight a significant gap between the frequency domains where real and fake audio features predominantly exist. Fake audio features are concentrated in higher frequency ranges compared to real audio. In most real-world applications, criminals aim to make fake audio sound real enough to deceive detectors for committing fraud. Therefore, focusing adversarial training on high frequencies effectively enhances the robustness of fake audio.

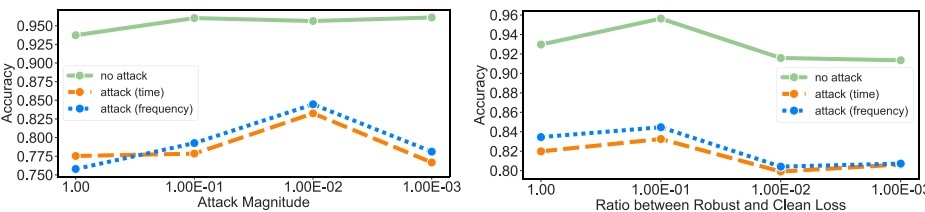

(a) Impact of Attack Magnitude $\epsilon$ on Detection Accuracy

(b) Impact of ratio $\gamma$ between robust and clean loss ($L_{robust}/L_{clean}$) on Detection Accuracy

Figure 7: Exploration of hyperparameters to control attacks, balancing accuracy and robustness. (We train Rawnet3 + RandAug + F-SAT on 10% of DeepFakeVoc-HQ)

**Balance Between Clean and Attacked Data:** Figure 7a demonstrates that F-SAT achieves optimal performance with an attack magnitude of $\epsilon = 0.01$, maintaining high accuracy on both attacked and clean data. Figure 7b reveals that a ratio of 0.1 between robust and clean loss, ($L_{robust}/L_{clean}$), results in the highest overall accuracy. The results show that excessive focus on attacked data does not consistently boost robustness and may decrease clean data accuracy. Similarly, an undue emphasis on clean data does not reliably enhance accuracy and can weaken robustness. Thus, striking a balance between clean and attacked samples is critical for optimal model performance.

**Additional Analysis of the F-SAT:** The fundamental rationale for our algorithm lies in the dual nature of high-frequency features: while they are crucial for identifying deepfake audio, they are also susceptible to adversarial attacks. In contrast, low-frequency features, though more robust, do not provide sufficient information on their own to train an effective detector. Therefore, simply using a lowpass filter to eliminate all high-frequency features is not a feasible strategy; instead, a powerful and resilient detector must retain and protect these high-frequency features.

Additionally, our F-SAT approach specifically targets the magnitude component of audio signals while preserving the phase. This focused strategy significantly bolsters the effectiveness of our method. To substantiate this approach, we conducted frequency-selective adversarial training on the high-frequency phase component, which led to a substantial decline in accuracy on unattacked data. This outcome suggests that the model may overfit to artifacts introduced by phase perturbations, thereby diminishing its generalization capacity.

## 6 CONCLUSION

In our study, we introduce a dataset *DeepFakeVoc-HQ* that addresses diversity and quality issues in prior datasets, and provide a taxonomy to explore common audio corruptions and attacks. We find that leading AI voice detection models depend on vulnerable high-frequency features. This discovery leads us to develop F-SAT, a targeted adversarial training method that focuses on high-frequency components while while preserving the integrity of low-frequency features. Our approach effectively maintains accuracy on unattacked data and enhances robustness against various attacks. These results pioneer robust training for detecting fake audio for the first time, opening up a new direction for identifying such threats.

ACKNOWLEDGMENTS

This work was supported in part by multiple Google Cyber NYC awards, Columbia SEAS/EVPR Stimulus award, and Columbia SEAS-KFAI Generative AI and Public Discourse Research award.

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

# A APPENDIX

## A.1 LEGAL PERMISSIONS FOR BENCHMARK DATASET

Our benchmark dataset is sourced from three main platforms: YouTube, commercial software, and open-source projects from GitHub. For both commercial software and open-source projects, we have reviewed their respective policies and confirmed that they permit the sharing of generated data for non-commercial use.

To address concerns regarding YouTube's terms of service and ethical considerations, we will not directly distribute any content sourced from YouTube. Instead, we will provide only metadata (video ID, start time, and end time), ensuring that researchers can access the content through YouTube's official interface in compliance with the platform's policies.

## A.2 COMPARISON OF DATASETS

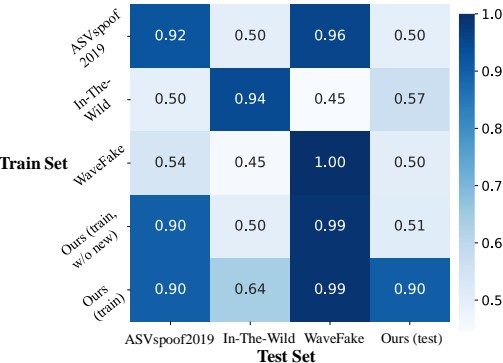

Figure 8: Performance of the RawNet3 baseline model on various datasets. 'Ours (train, w/o new)' represents our training dataset after removing all high-quality deepfake samples.

## A.3 EXPLANATION OF SOME TYPES OF CORRUPTIONS

**Air Absorption:** Air absorption refers to the phenomenon where high-frequency sound waves are more strongly attenuated by air molecules. As sound travels through the air, it loses energy, particularly at higher frequencies, due to air viscosity and thermal conduction.

**Room Simulator:** A room simulator is a digital tool that emulates the acoustics of different rooms or spaces. It includes parameters such as room size, wall materials, and shape, which affect how sound reflects and diffuses.

**Peaking:** Peaking refers to adjusting a specific range of frequencies around a central frequency. It is commonly used in parametric equalizers.

**Aliasing:** Aliasing occurs when high-frequency components are sampled below the Nyquist rate, causing them to appear as lower frequencies. This can be represented using the Nyquist theorem:

$$f_{\text{sample}} > 2 \cdot f_{\text{max}},$$

where $f_{\text{sample}}$ is the sampling frequency, and $f_{\text{max}}$ is the maximum frequency of the signal. When this condition is violated, the signal is folded back into the lower frequencies, creating aliasing artifacts.

**Bit-Crush:** Bit-crushing reduces the bit depth of an audio signal, causing quantization errors.

**Tanh Distortion:** Tanh distortion is a form of soft clipping achieved using the hyperbolic tangent function. The output $y$ is given by:

$$y = \tanh(kx),$$

where $x$ is the input signal, and $k$ controls the amount of distortion. As $k$ increases, the function approximates a hard clipping effect.

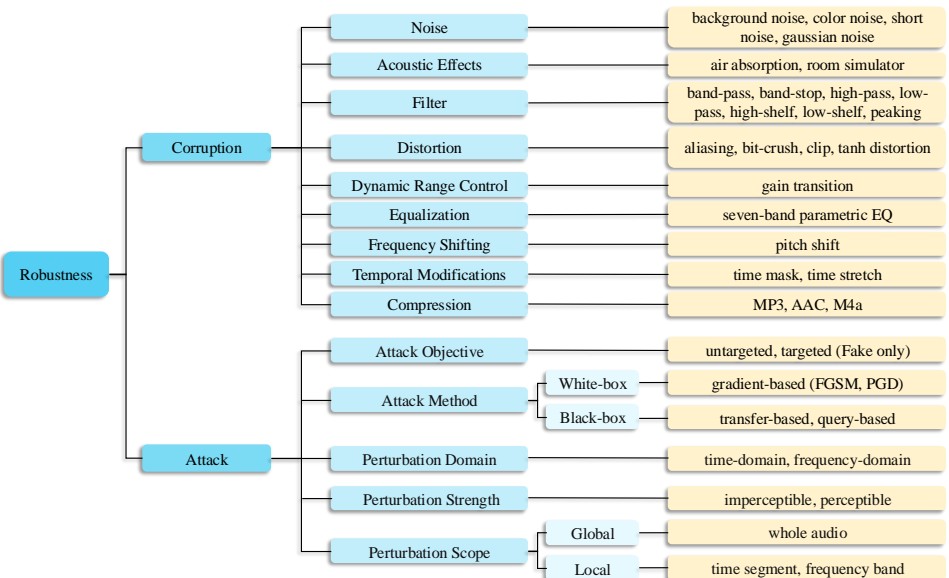

Figure 9: Overview of various corruption types and adversarial attack strategies affecting audio robustness. The diagram categorizes different forms of corruptions (e.g., noise, filtering, distortion) and adversarial attacks (e.g., white-box, black-box) based on their methods, objectives, and scope of perturbation. This framework outlines the challenges in ensuring the robustness of audio systems against both environmental corruption and intentional adversarial manipulation.

**Gain Transition:** Gain transition refers to smoothly adjusting the amplitude over time.

**Seven-Band Parametric EQ:** A seven-band parametric EQ provides independent control over seven frequency bands. Each band can be adjusted using three parameters: center frequency ($f_0$), gain ($G$), and bandwidth ($Q$). The overall transfer function is a combination of individual band filters:

$$H(f) = \prod_{i=1}^{7} H_i(f),$$

where $H_i(f)$ represents the frequency response of the $i$-th band.

**Time Mask:** Time masking involves temporarily silencing or removing a segment of audio.

## A.4 RANDAUGMENT FOR AUDIO

```python
audio_transforms = [
    'background_noise', 'color_noise', 'short_noise', 'gaussian_noise', 'air_absorption', 'room_simulator', 'band_pass',
    'band_stop', 'high_pass', 'low_pass', 'high_shelf', 'low_shelf', 'peaking', 'aliasing', 'bit_crush', 'clip',
    'tanh_distortion', 'gain_transition', 'seven_band_parametric_EQ', 'pitch_shift', 'time_mask', 'time_stretch'
]
def rand_augment_audio(sample, N, p):
    """ Apply random augmentations to an audio sample.
    Args:
        sample: An audio sample    N: Number of augmentations to apply    p: Probability of applying each augmentation
    Returns:
        An augmented audio sample
    """

    operations = np.random.choice(audio_transforms, N)
    return operations(sample) if random.random() < p else sample
```

Figure 10: Python code for RandAugment for audio

Details of RandAugment for Audio are shown in Figure 10. In our experiments, we set $N = 1$ and $p = 0.9$.

## A.5 ROBUSTNESS TO COMPRESSION

To evaluate the robustness of our detection model to compression, we tested two lossy formats: MP3 and AAC. The evaluation utilized RawNet3 combined with RandAug and F-SAT. As shown in the table 5, both MP3 and AAC compression had minimal impact on detection accuracy.

| Format | Real | Fake | Avg |
|---|---|---|---|
| Origin (90% wav + 10% mp3) | 97.5% | 96.4% | 97.0% |
| MP3 | 97.5% | 95.6% | 96.6% |
| ACC | 96.9% | 96.6% | 96.8% |

Table 5: Detection Results for Compressed Audio Formats

## A.6 COMPREHENSIVE ANALYSIS OF ALL ATTACK TYPES

Table 6 presents the detailed adversarial attack results for RawNet3 and our method under various conditions. It includes both white-box and black-box approaches and examines the impacts of attacks in both the time and frequency domains, detailing the different attack hyperparameters used.

| Domain | Iters | Restart | Source | RawNet3 | | RawNet3+RandAug | | RawNet3+RandAug+F-SAT | |
|---|---|---|---|---|---|---|---|---|---|
| | | | | **Real Acc** | **Fake Acc** | **Real Acc** | **Fake Acc** | **Real Acc** | **Fake Acc** |
| No Attack | - | - | - | 96.1% | 75.9% | **97.6%** | 94.7% | 97.5% | **96.4%** |
| Time | 2 | 1 | A | **97.9%** | 3.50% | 74.7% | 66.1% | 90.2% | **88.0%** |
| | 5 | 1 | A | **90.6%** | 2.10% | 34.4% | 42.4% | 66.7% | **80.9%** |
| | 5 | 2 | A | **90.2%** | 1.70% | 33.2% | 39.4% | 65.4% | **80.3%** |
| | 2 | 1 | A' | **98.9%** | 8.10% | 79.8% | 69.0% | 92.9% | **89.2%** |
| | 5 | 1 | A' | **92.1%** | 2.90% | 43.7% | 52.4% | 74.4% | **84.0%** |
| | 5 | 2 | A' | **91.6%** | 3.20% | 42.1% | 50.3% | 73.2% | **83.1%** |
| | 2 | 1 | B | 100.0% | 17.3% | **98.7%** | 91.7% | 98.7% | **96.4%** |
| | 5 | 1 | B | 100.0% | 16.0% | **98.9%** | 92.1% | 98.7% | **96.3%** |
| | 5 | 2 | B | 100.0% | 16.2% | **98.9%** | 92.1% | 98.7% | **96.3%** |
| Frequency (0-8k Hz) | 2 | 1 | A | 65.0% | 5.7% | 38.5% | 42.9% | **87.0%** | **86.3%** |
| | 5 | 1 | A | 23.8% | 2.9% | 7.5% | 14.9% | **63.8%** | **69.4%** |
| | 5 | 2 | A | 23.0% | 2.7% | 7.3% | 15.2% | **63.7%** | **69.2%** |
| | 2 | 1 | A' | 77.3% | 14.9% | 50.6% | 55.1% | **90.2%** | **88.1%** |
| | 5 | 1 | A' | 31.6% | 4.92% | 14.9% | 27.6% | **72.7%** | **74.6%** |
| | 5 | 2 | A' | 23.8% | 2.4% | 7.8% | 16.0% | **63.8%** | **66.4%** |
| | 2 | 1 | B | 99.8% | 34.60% | 98.9% | 85.98% | **98.7%** | **96.8%** |
| | 5 | 1 | B | 99.8% | 29.5% | 98.9% | 85.3% | **98.7%** | **96.8%** |
| | 5 | 2 | B | **100.0%** | 31.9% | 98.9% | 84.9% | 98.7% | **96.7%** |
| Frequency (2-8k Hz) | 2 | 1 | A | 74.3% | 6.8% | 50.4% | 49.3% | **93.1%** | **89.5%** |
| | 5 | 1 | A | 34.0% | 4.29% | 12.9% | 22.9% | **83.5%** | **82.6%** |
| | 5 | 2 | A | 33.0% | 4.44% | 12.7% | 21.4% | **83.0%** | **82.1%** |
| | 2 | 1 | A' | 84.0% | 16.5% | 59.7% | 63.5% | **94.0%** | **89.8%** |
| | 5 | 1 | A' | 43.8% | 7.00% | 23.5% | 35.2% | **87.5%** | **85.1%** |
| | 5 | 2 | A' | 33.5% | 4.0% | 12.9% | 21.3% | **82.9%** | **80.3%** |
| | 2 | 1 | B | 99.8% | 33.3% | 98.9% | 84.2% | **98.6%** | **97.0%** |
| | 5 | 1 | B | **100.0%** | 28.4% | 98.4% | 84.7% | 98.7% | **95.9%** |
| | 5 | 2 | B | 99.8% | 31.0% | 99.1% | 84.2% | **98.6%** | **96.0%** |
| Frequency (4-8k Hz) | 2 | 1 | A | 87.2% | 8.5% | 72.9% | 61.3% | **96.7%** | **94.6%** |
| | 5 | 1 | A | 54.9% | 7.78% | 35.4% | 36.3% | **96.2%** | **93.4%** |
| | 5 | 2 | A | 54.4% | 6.83% | 35.1% | 36.3% | **95.7%** | **93.2%** |
| | 2 | 1 | A' | 91.4% | 18.3% | 80.0% | 71.9% | **97.1%** | **95.6%** |
| | 5 | 1 | A' | 64.9% | 10.5% | 47.3% | 50.6% | **96.5%** | **94.5%** |
| | 5 | 2 | A' | 54.6% | 7.1% | 36.0% | 36.3% | **96.0%** | **93.4%** |
| | 2 | 1 | B | 99.7% | 34.4% | 99.0% | 88.0% | **98.0%** | **96.3%** |
| | 5 | 1 | B | 99.7% | 29.2% | 99.0% | 87.5% | **98.0%** | **96.3%** |
| | 5 | 2 | B | 99.7% | 32.5% | 99.2% | 87.8% | **98.0%** | **96.3%** |
| Frequency (6-8k Hz) | 2 | 1 | A | 95.2% | 12.3% | 90.1% | 76.7% | **96.5%** | **95.6%** |
| | 5 | 1 | A | 83.8% | 16.0% | 70.6% | 65.6% | **95.7%** | **95.3%** |
| | 5 | 2 | A | 83.7% | 15.7% | 70.5% | 65.0% | **95.9%** | **94.9%** |
| | 2 | 1 | A' | 95.7% | 21.9% | 92.7% | 80.4% | **96.7%** | **95.9%** |
| | 5 | 1 | A' | 86.8% | 16.8% | 78.4% | 70.4% | **96.0%** | **95.9%** |
| | 5 | 2 | A' | 83.2% | 15.9% | 70.5% | 63.4% | **95.9%** | **95.8%** |
| | 2 | 1 | B | 99.7% | 38.3% | 98.9% | 89.1% | **97.0%** | **96.4%** |
| | 5 | 1 | B | 99.7% | 32.2% | 98.9% | 88.5% | **97.1%** | **96.4%** |
| | 5 | 2 | B | 99.7% | 36.8% | 99.0% | 88.5% | **97.3%** | **96.4%** |

Table 6: Adversarial Attack Results on RawNet3 and Its Variants under Various Conditions. For the attack scenarios, we include both white-box and black-box approaches: $A$ represents tests with the same model and identical weights, while $A'$ indicates the same model but with different weights, and $B$ denotes tests on a completely different model. For attacks in the time domain, we use $\epsilon = 10^{-4}$ and $\alpha = 4 \cdot 10^{-5}$. For attacks in the frequency domain, the parameters are $\epsilon = 10^{-3}$ and $\alpha = 4 \cdot 10^{-4}$.

## A.7  DATASET DETAILS

| Fake Source | Total Duration (Hours) | Audio Count | Mean Duration (Seconds) |
|---|---|---|---|
| VCTK | 14.1 | 12.0k | 4.2 |
| Librispeech (Train) | 961.1 | 281.2k | 12.3 |
| In-The-Wilds | 14.6 | 9.3k | 5.7 |
| ASRspoof2019 (LA) | 11.9 | 12.5k | 3.4 |
| Voxceleb1 | 340.4 | 148.6k | 8.2 |
| Audioset (Narration) | 50.1 | 12.2k | 14.8 |

Table 7: Summary of fake audio sources data

| Fake Source | Total Duration (Hours) | Audio Count | Mean Duration (Seconds) |
|---|---|---|---|
| Metavoice | 189.1 | 61.7k | 11.0 |
| StyleTTS-v2 | 186.6 | 61.6k | 10.9 |
| XTTS-v2 | 175.5 | 61.8k | 10.2 |
| VoiceCraft | 119.9 | 59.4k | 7.3 |
| Whisperspeech | 155.2 | 61.9k | 9.0 |
| Vokan-TTS | 161.7 | 61.6k | 9.4 |
| Elevenlabs | 3.3 | 3.2k | 3.7 |
| ASRspoof2019 (LA) | 97.8 | 109.0k | 3.2 |
| Wavefake (English) | 198.7 | 117.9k | 6.1 |

Table 8: Summary of fake audio sources data

| | VCTK Speaker_id: p244 | | | | In-the-Wild Speaker: Alan Watts | | | |
|---|---|---|---|---|---|---|---|---|
| **Model** | **Ovrl MOS** | **Sig MOS** | **Bak MOS** | **P808 MOS** | **Ovrl MOS** | **Sig MOS** | **Bak MOS** | **P808 MOS** |
| Real refer | 3.26 | 3.56 | 4.04 | 3.61 | 3.02 | 3.40 | 3.74 | 3.57 |
| metavoice | **3.29** | **3.58** | 4.05 | 3.63 | 3.15 | 3.52 | 3.88 | 3.55 |
| StyleTTS v2 | 3.28 | 3.56 | **4.08** | **3.87** | **3.28** | **3.57** | **4.06** | **3.83** |
| XTTS v2 | 3.13 | 3.41 | 4.00 | 3.78 | 3.11 | 3.41 | 3.98 | 3.70 |
| VoiceCraft | 3.16 | 3.51 | 3.94 | 3.61 | 3.01 | 3.34 | 3.80 | 3.43 |
| Whisperspeech | 3.28 | 3.56 | 4.07 | 3.82 | 3.15 | 3.44 | 3.99 | 3.59 |
| Vokan-TTS | 3.23 | 3.55 | 4.01 | 3.71 | 2.94 | 3.39 | 3.66 | 3.60 |

Table 9: MOS Scores for Various TTS Models

Table 7 presents the real audio data we utilized from previously published datasets. For generating deepfake audio for the training set, we employ models such as XTTS v2, StyleTTS v2, Metavoice, Whisperspeech, Vokan-TTS, VoiceCraft, and Elevenlabs. Additionally, for the test set, we use Cosyvoice, PlayHT 2.0, Resemble, LOVO AI, and Lipsynthesis to create deepfake voices. We introduce post-processing augmentations to generate noisy deepfakes. Four real datasets—VCTK (12.0k), Librispeech-clean-100 (28.5k), Audioset (narration) (12.2k), and In-The-Wild (real parts: 9.3k)—are utilized to generate deepfake voices for the training set. Details of our generated deepfake audio and references to previous public deepfake audio datasets are presented in Table 8. Additionally, we employed DNSMOS (Reddy et al., 2021) to quantitatively measure the synthetic speech quality across these models, using a scale from 1 to 5, where higher values indicate better quality. As demonstrated in Table 9, Metavoice and StyleTTS outperform other AI voice synthesis models.

## A.8 F-SAT'S TRAINING EFFICIENCY

F-SAT's training efficiency is influenced by hyperparameters such as attack iterations and restart counts, which identify the worst-case perturbation. Drawing on insights from "Fast Is Better Than Free: Revisiting Adversarial Training," we optimized these parameters by setting restarts to one and attack iterations to one or two, while employing a larger attack magnitude to enhance robustness. Training time is shown in the Table 10. Although F-SAT requires longer training times, it improves accuracy by an average of 9% on original data and 43% on attacked data compared to Standard Adversarial Training. We should not compromise accuracy merely to accelerate training

| Description | w/o Adversarial Training | Standard Adversarial Training | F-SAT |
|---|---|---|---|
| Training Duration (Days) | 2 | 4.5 | 8 |
| Number of Epochs | 15 | 15 | 15 |
| Hardware Used | A100 GPU | A100 GPU | A100 GPU |

Table 10: Training time comparasion

### A.8.1 TRAINING HYPERPARAMETER

Here are the Training hyperparameter of F-SAT for Table 3:

**Training Hyperparameters**

- **Learning Rate (lr):** $1 \times 10^{-5}$
- **Epochs:** 15
- **Batch Size (bs):** 16
- **Optimizer:** adam
- **Augmentation Number (aug_num):** 1 or 2
- **Augmentation Probability (aug_prob):** 0.9

**LR Scheduler (Warmup Cosine)**

- **Warm-up Epochs:** 1
- **Warm-up LR:** $1 \times 10^{-6}$
- **Minimum LR:** $1 \times 10^{-7}$

**Attack Hyperparameters**

- **Attack Type:** $l_\infty$
- **Epsilon:** 0.005, **Alpha:** 0.002
- **Gamma (control ratio of clean loss and robust loss):** 0.1
- **Attack Iterations:** 2
- **Restarts:** 1
- **Frequency Range:** 4-8k Hz

**Mixup Hyperparameters**

- **Mixup Alpha:** 0.5