# OpenReview forum: "I Can Hear You: Selective Robust Training for Deepfake Audio Detection"
_ICLR.cc/2025/Conference — ICLR 2025 Poster_

### Official Review · Reviewer_NjHP · 2024-10-25

**Soundness:** 3
**Presentation:** 2
**Contribution:** 2
**Rating:** 6
**Confidence:** 3

**Summary:**

The paper "I CAN HEAR YOU: SELECTIVE ROBUST TRAINING FOR DEEPFAKE AUDIO DETECTION"
introduces the DeepFakeVox-HQ data set, which contains audio from 14 sources.
In addition to the dataset, the authors introduce Frequency-Selective Adversarial Training (F-SAT), a training method that focuses on the high-frequency part of the spectrum. In addition to FSAT, this paper evaluates robustness concerning various input perturbations.

**Strengths:**

- The paper tackles a significant problem.
- The related work is well-researched and described.
- The adversarial attack perspective is interesting.
- Authors ensure their results are up to date, combining existing datasets with samples from commercial models.

**Weaknesses:**

- Traditional compression algorithms like MP3 remove high-frequency content; according to line 84, FSAT focuses on this part of the spectrum.
- If I understand correctly, compression is not part of the corruption set, as shown in Figure 7. Including compression would have been important for real-world applicability.
- Data-set details like the length in hours or training hyperparameters like the learning rate are missing.

**Questions:**

- Which model was used to create the plot on the left side in Figure 2?
- How many hours of speech does the dataset encompass exactly? How long are the samples?
- Are the samples aligned? Do all models synthesize speech using the same input sentences?
- Is it possible to add a data sheet that outlines the exact sources and utterance lengths per source?
- Are the WaveFake test samples also part of the DeefFakeVox-HQ test set?
- WaveFake contains Japanese language JSUT samples.
    - Are these part of the dataset?
    - Should the Caption of Table 1 make this explicit? Since WaveFake is listed as
     English-language data set, I assume JSUT is not considered a part of WaveFake in this paper.
    - Do the Utterance numbers in table one exclude JSUT?
    - If yes, should this be mentioned somewhere else?

- Is it possible to include leading works from the audio classification world, like the Audio Spectrogram Transformer (AST)[1], in the evaluations? Related work [2] found it to perform well on the WaveFake-dataset. It would be interesting if it also outperforms other methods on DeepFakeVox-HQ.

- The Wavefake paper [3] trains with binary settings with fake audio from a single source and measures generalization. Training on which source network led to the numbers in Table 2? Are the numbers comparable to the related work?

- Which software libraries have been used to implement this project?

- Which hyperparameters underpin the network training?

Related work:
[1] AST: Audio Spectrogram Transformer, https://arxiv.org/pdf/2104.01778,
[2] Towards generalizing deep-audio fake detection networks, https://arxiv.org/pdf/2305.13033,
[3] WaveFake: A Data Set to Facilitate Audio Deepfake Detection, https://arxiv.org/abs/2111.02813

---

> ### Author Response · Authors · 2024-11-25
> **Thank you for your your thoughtful review and helpful experiment suggestions!**
>
> We appreciate the reviewer's positive feedback on our paper. It is encouraging to see the recognition of our tackling a significant problem, the interest in our adversarial attack perspective, and our efforts to keep results current by integrating samples from commercial models. We discuss these contributions and address reviewer suggestions in the sections below.
>
> **W1 - MP3 compression removes high-frequency content, which may impact the effectiveness of F-SAT**
>
> Traditional compression algorithms like MP3 typically remove frequencies above 16 kHz. However, in our experiment, the audio is resampled to 16,000 Hz before being input to the model, capping the maximum representable frequency at 8,000 Hz (per the Nyquist theorem). Thus, MP3’s high-frequency removal has no impact on our data, as it already falls below this range.
>
> **W2 - Robustness to compression**
>
> Thank you for your suggestion. We have added results on compressed audio in the appendix. To evaluate the robustness of our detection model to compression, we tested two lossy formats: MP3 and AAC. The evaluation utilized RawNet3 combined with RandAug and F-SAT. As shown in the table, both MP3 and AAC compression had minimal impact on detection accuracy.
>
> | Format                     | Real    | Fake    | Avg     |
> |----------------------------|---------|---------|---------|
> | Origin (90% wav + 10% mp3) | 97.50%  | 98.40%  | 98.00%  |
> | MP3                        | 97.50%  | 97.60%  | 97.60%  |
> | ACC                        | 96.90%  | 98.60%  | 97.80%  |
>
> **W3 - Dataset details like the length in hours or training hyperparameters like the learning rate are missing**
>
> Thank you for bringing this up. We have added dataset details and training hyperparameters in the revised version.
> ##### Training Hyperparameters for RawNet3 on DeepfakeVox
> - **Learning Rate (lr):** `1e-5`
> - **Epochs:** `15`
> - **Batch Size (bs):** `16`
> - **Optimizer:** `adam`
> - **Augmentation Number (aug_num):** `1` or `2`
> - **Augmentation Probability (aug_prob):** `0.9`
>
> ##### LR Scheduler (Warmup Cosine)
> - **Warm-up Epochs:** `1`
> - **Warm-up LR:** `1e-6`
> - **Minimum LR:** `1e-7`
>
> ##### Attack Hyperparameters
> - **Attack Type:** `l_inf`
> - **Epsilon:** `0.005`, **Alpha:** `0.002`
> - **Gamma (control ratio of clean loss and robust loss):** `0.1`
> - **Attack Iterations:** `2`
> - **Restarts:** `1`
>
> ##### Mixup Hyperparameters
> - **Mixup Alpha:** `0.5`
>
> #### Dataset Details
> For the training set, we used six open-source models to generate data from four datasets: VCTK (12k samples), LibriSpeech-clean-100 (28k samples), AudioSet (narration) (12k samples), and In-The-Wild (real parts: 9k samples), resulting in a dataset nearly six times larger than the real samples and evenly distributed across these sources. Additionally, we used one commercial model, ElevenLabs, to generate 2,500 samples. The total duration is detailed below.
>
> |                | MetaVoice | StyleTTS v2 | XTTS v2 | VoiceCraft | Whisperspeech | Vokan-TTS | Elevenlabs |
> |----------------|-----------|-------------|---------|------------|---------------|-----------|------------|
> | Duration (hrs) | 189.1     | 186.6       |175.5   | 119.9       | 155.2         | 161.7     | 3.3        |
>
> We also combined our generated data with existing public datasets to enhance its diversity. For the test set, we included 14 different fake sources, with approximately 200 samples each.
>
> **Q1 - Which model was used to create the plot on the left side in Figure 2**
>
> RawNet3 Model (without F-SAT)
>
> **Q2 - How many hours of speech does the dataset encompass exactly? How long are the samples?**
>
> The total duration of our dataset is approximately 2700 hours, comprising 1400 hours of real audio and 1300 hours of fake audio. The fake audio includes 300 hours from a previous dataset and 1000 hours we generated. Sample lengths vary across subdatasets, ranging from 4 to 15 seconds. Specifically, for VCTK and the corresponding fake audio, the average duration is about 4 seconds per sample. In contrast, for AudioSet (narration), the average duration extends to around 15 seconds per sample.
>
>
> **Q3 - Are the samples aligned? Do all models synthesize speech using the same input sentences?**
>
> For the high-quality deepfake samples we generated, alignment is ensured. All models synthesize speech from the same input sentences.

---

> ### Author Response · Authors · 2024-11-25
> **Continue**
>
> **Q4 - Add a data sheet that outlines the exact sources and utterance lengths per source**
>
> Thanks for you suggestions. We have added a data sheet that outlines the exact sources and utterance lengths per source in revised version.
>
> | Real Source                 | VCTK   | Librispeech(Train) | In-The-Wilds | ASRspoof2019(LA) | Voxceleb1 | Audioset(Narration) |
> |------------------------|--------|----------------------|--------------|-----------------------|-----------|---------------------|
> | Total Duration (Hours) | 14.1   | 961.1                | 14.6         | 11.9                  | 340.4     | 50.1                |
> | Audio Count (k)        | 12.0k  | 281.2k               | 9.3k         | 12.5k                 | 148.6k    | 12.2k               |
> | Mean Duration (Seconds)| 4.2    | 12.3                 | 5.7          | 3.4                   | 8.2       | 14.8                |
>
>
> | Fake Source          | Metavoice | StyleTTS-v2 | XTTS-v2 | VoiceCraft | Whisperspeech | Vokan-TTS | Elevenlabs | ASRspoof2019(LA) | Wavefake(English) |
> |--------------------------|-----------|-------------|---------|------------|---------------|-----------|------------|-----------------|--------------------|
> | Total Duration (Hours)   | 189.1     | 186.6       | 175.5   | 119.9      | 155.2         | 161.7     | 3.3        | 97.8            | 198.7              |
> | Audio Count              | 61.7k     | 61.6k       | 61.8k   | 59.4k      | 61.9k         | 61.6k     | 3.2k       | 109.0k          | 117.9             |
> | Mean Duration (Seconds)  | 11.0      | 10.9        | 10.2    | 7.3        | 9.0           | 9.4       | 3.7        | 3.2             | 6.1                |
>
> **Q5 - Are the WaveFake test samples also part of the DeefFakeVox-HQ test set?**
>
> No, WaveFake utilizes six AI synthesis models: MelGAN, ParallelWaveGAN, Multi-band MelGAN, Full-band MelGAN, HiFi-GAN, and WaveGlow, none of which are covered in our test set.
>
> **Q6 - Clarify for WaveFake Dataset**
>
> We apologize for any confusion caused; our experiment utilized only the English portion of the WaveFake dataset. We have updated our paper accordingly, revising Table 1 to indicate that WaveFake includes both English and Japanese languages. Additionally, in Section 5.1, we have clearly specified that we used only the English component of the dataset in revised version.
>
> **Q7 - Audio Spectrogram Transformer performance on DeefFakeVox-HQ**
>
> Audio Spectrogram Transformer does not perform as well as RawNet3 on DeepFakeVox-HQ. We use the same training hyperparameters (learning rate schedule, optimizer, batch size, etc.) and same augmentation.
>
> | Model              | Real   | Fake   | Avg    |
> |--------------------|--------|--------|--------|
> | AST + Randaug      | 99.4%  | 78.0%  | 88.7%  |
> | RawNet3 + Randaug  | 97.6%  | 97.0%  | 97.3%  |
>
> **Q8 - Setting for training Wavefake in Table 2**
>
> Sorry for the confusion, the training settings for the WaveFake dataset in our paper differ from those in the original paper to maintain consistency across all datasets. We used all available sources and divided them into training, validation, and testing sets with a ratio of 7:1.5:1.5.
>
> **Q9 - Which software libraries have been used to implement this project?**
>
> For the detection model, we used standard libraries such as Torch, Torchaudio, Scikit-learn, and Librosa. Detailed information on each package and its version is included in the supplementary file env.txt, which lists all dependencies. The software libraries used by TTS (Text-to-Speech) and VC (Voice Conversion) models to generate deepfake audio, however, vary. If accepted, we will open-source our code and data and provide a comprehensive README.

---

> ### Comment · Reviewer_NjHP · 2024-11-25
> **Thank you for answering my questions**
>
> Dear authors,
>
> My last remaining concern is with Q8 (Table 2). Is it possible to explicitly state that the results are not comparable to the original wavefake paper? Furthermore, Wavefake does not exclusively contain LJSpeech samples. It also has a JSUT part. Is it possible to fix the caption?

---

> > ### Author Response · Authors · 2024-11-25
> > **Thanks for your suggestion**
> >
> > Dear Reviewer:
> > Thank you for your guidance. We have revised the caption of Table 2 to clearly state that the results are not directly comparable to those in the original WaveFake paper. Additionally, Section 5.1, “Introduction to WaveFake,” has been thoroughly updated to detail the differences in experimental settings and dataset utilization.

---

> > > ### Comment · Reviewer_NjHP · 2024-11-26
> > > **Thank you for responding.**
> > >
> > > Thank you for updating the paper. It was essential to avoid conflicting numbers in the future and make the change in the experimental setting explicit. That way, future work won't have to make impossible comparisons. I wish to maintain my original score.

---

> > > > ### Comment · Reviewer_NjHP · 2024-11-26
> > > > **A last concern**
> > > >
> > > > Dear authors,
> > > > I have had another look at lines 359-362 of the paper
> > > > `` Our method
> > > > achieved state-of-the-art results across all three benchmarks. Compared with RawNet3, our method
> > > > shows improvements of 7.7% points on DeepFakeVox-HQ (test), 8.4% points on ASVspoof2019,
> > > > and 0.1% points on WaveFake. ``
> > > >
> > > > I am not sure the claim on WaveFake is valid since the experimental setup is different. Furthermore, RawNet3 did not produce good results in the WaveFake study. Would it be possible to reproduce the original WaveFake setup and compare FSAT to the GMM that performed best in the original study? I think a fair comparison would be critical here. If that is not possible in the remaining time, it's probably better not to make claims regarding performance on WaveFake.

---

> > > > > ### Author Response · Authors · 2024-11-26
> > > > > **Thank you for sharing your concern**
> > > > >
> > > > > Thank you for sharing your concern. To address potential misunderstandings, we have removed the results related to WaveFake. We would also like to clarify that the key contribution of our work lies in the robust training method applied to the detection model, rather than the detection model itself. In Table 2, our objective is to demonstrate that our adversarial training method maintains accuracy on the original data (RawNet3 vs. ours). The comparison of RawNet3 with other baseline models was conducted solely to determine the most suitable detection model for our training dataset. That's the reason why we did not replicate the exact experimental settings or compare our results directly with those in the original paper.

---

> > > > > > ### Comment · Reviewer_NjHP · 2024-11-27
> > > > > >
> > > > > > Okay, that works for me. I maintain my score.

---

### Official Review · Reviewer_erKc · 2024-11-02

**Soundness:** 4
**Presentation:** 4
**Contribution:** 3
**Rating:** 6
**Confidence:** 3

**Summary:**

Glad to review the paper.
This paper proposes a novel method, F-SAT for deepfake audio detection.
The topic of this work is promising, and the paper is easy to follow.
I believe this work has reference values to domain-related researchers.

**Strengths:**

Three main contributions involved in this work include (1) a carefully organized dataset, (2) a deepfake detection method, and (3) the ability against adversarial attacks (with the setting focusing on high-frequency signals).
In general, the contributions of this work are multi-fold.

**Weaknesses:**

My major concern is whether the contributions (or advantages) of this work are over-claimed.
Regarding the dataset, although it is well organized and processed, the samples are generated using existing approaches, thus, "the largest" is not a significant contribution.
Regarding generalization, as in Table 2, the significantly superior results of the proposed method are achieved on the self-organized dataset, DeepFake Vox-HQ. However, as the author introduced in Section 3, there are overlapped synthesis methods between training and testing data in this group of results (as in Figure 6). Thus the results in DeepFake Vox-HQ can not indicate out-of-distribution generalization.
Regarding enhancing robustness, in the last paragraph of the related section, the referenced solutions were published in 2019,2018 and 2018, I am not sure whether any recent works focus on the adversarial issue, that should be discussed or compared.

**Questions:**

My main concern is about the generalization and robustness issues listed above.
I will consider changing my score based on the author's responses and other reviewers.

---

> ### Author Response · Authors · 2024-11-25
> **Thank you for your thoughtful review and useful suggestions**
>
> We appreciate the reviewer's insightful feedback on our work. Key strengths highlighted include our comprehensive dataset, advanced deepfake detection method, and robust defense against high-frequency adversarial attacks. These contributions collectively push forward the boundaries in deepfake detection research. Further details and additional insights are discussed in the subsequent sections.
>
> **W - contribution on our dataset**
>
> Thank you for your suggestions. We are glad that the reviewer recognize our dataset to be well organized and processed. As suggested, we will remove the claim of “largest” in the revised version. Our dataset comprehensively summarizes existing TTS and VC models, evaluating over 30 AI-synthesis models proposed within the last 2-3 years, including both open-source and commercial models. We utilized 14 of these models, which we believe are significant to the community and merit publication.
>
> **W - "Compared with other defense method**
>
> Thank you for the suggestion. We have included a discussion of the paper ‘High-frequency Adversarial Defense for Speech and Audio’ (2021) in the related work section and present comparative results in the experiments section. We appreciate Reviewer EGHT's recommendation for this comparison.
>
> We compared our method against MAD smoothing[1], as suggested in the paper, and also included Gaussian smoothing[3], and adversarial training in the time domain[2], which are used as baselines in the reference paper. We employed the same parameters as those in [1]. The results of these comparisons are outlined below. As the table shows, our method outperforms all other methods.
>
> | Approach               | Ori Real | Ori Fake | Ori Avg | Att(T) Real | Att(T) Fake | Att(T) Avg | Att(F) Real | Att(F) Fake | Att(F) Avg |
> |------------------------|----------|----------|---------|-------------|-------------|------------|-------------|-------------|------------|
> | RawNet3+RandAug        | **97.6%**    | 97.0%    | 97.3%   | 74.7%       | 66.0%       | 70.4%      | 63.0%       | 62.4%       | 62.7%      |
> | +AT(Time)              | 94.7%    | 83.1%    | 88.9%   | 87.1%       | 12.4%       | 49.8%      | 66.5%       | 16.5%       | 41.5%      |
> | +Gaussian smooth       | 57.8%    | 49.4%    | 53.6%   | 53.2%       | 51.3%       | 52.2%      | 46.2%       | 51.4%       | 48.8%      |
> | +MAD smoothing         | 96.2%    | 91.4%    | 93.8%   | 60.2%       | 59.0%       | 59.6%      | 56.3%       | 57.4%       | 56.8%      |
> | +F-SAT (Ours)          | 97.5%    | **98.4%**    | **98.0%**   | **90.2%**       | **87.0%**       | **88.6%**      | **93.3%**       | **92.8%**       | **93.1%**      |
>
> We also found that robust deepfake voice detection lacks enough related work in recent years. However, with GenAI making the deepfake generation increasingly realistic and widespread, it is urgent and crucial to design tools to detect them in a robust way. As far as we know, Our work is the first to tackle deepfake audio detection in a real-world setup after the boom of recent GenAI. Our work will be important in publishing and mitigating the urgent risk from generated audio.
>
> [1] Olivier, Raphael, Bhiksha Raj, and Muhammad Shah. "High-frequency adversarial defense for speech and audio." ICASSP 2021-2021 IEEE International Conference on Acoustics, Speech and Signal Processing (ICASSP). IEEE, 2021
>
> [2] Aleksander Madry, Aleksandar Makelov, Ludwig Schmidt, Dimitris Tsipras, and Adrian Vladu, “Towards deep learning models resistant to adversarial attacks,” international conference on learning representations, 2018.
>
> [3] Jeremy Cohen, Elan Rosenfeld, and Zico Kolter, “Certified adversarial robustness via randomized smoothing,” in Proceed- ings of the 36th International Conference on Machine Learning, Kamalika Chaudhuri and Ruslan Salakhutdinov, Eds. 09– 15 Jun 2019, vol. 97 of Proceedings of Machine Learning Research, pp. 1310–1320, PMLR.
>
> **W - DeepFakeVox-HQ can not indicate out-of-distribution generalization**
>
> Apologies for any confusion. The DeepFakeVox-HQ test set featured in our paper comprises entirely out-of-distribution (OOD) samples. As noted in the caption for Table 2, our experiments include seven fake sources not utilized in the training set. This also applies to real samples from recent YouTube videos, which are OOD as well.
>
> For future research, we provide additional test samples from seven fake sources that were included in the training set, although these were not used in the current study. We have clarified this at the beginning of the experiments section to prevent any further confusion.

---

> > ### Comment · Reviewer_erKc · 2024-11-26
> > **Thanks for the response.**
> >
> > Thank the authors for your responses, which have addressed my concerns to some extent, and I have raised my score to 6.
> >
> > Regarding the generalization issue evaluated through a self-built dataset, I suggest the authors to provide additional explanations in the final version, if the work could be accepted.

---

### Official Review · Reviewer_ijUY · 2024-11-02

**Soundness:** 3
**Presentation:** 3
**Contribution:** 3
**Rating:** 8
**Confidence:** 5

**Summary:**

This paper addresses the challenge of deepfake audio detection, presenting two major contributions: (1) the creation of DeepFakeVox-HQ, the largest and most diverse public dataset for deepfake audio detection, which enables realistic testing conditions and exposes limitations in existing models, and (2) the introduction of Frequency-Selective Adversarial Training (F-SAT), a novel approach that improves detection robustness by focusing on high-frequency audio components. The work is well-written and logically structured, making complex concepts accessible, and holds significant potential for advancing the robustness and reliability of deepfake audio detection models.

**Strengths:**

1.DeepFakeVox-HQ stands out as a substantial addition to the field, with over 1.3 million samples, including 270,000 high-quality deepfake samples from 14 sources. This dataset addresses the limitations of existing datasets in diversity and scale, making it a valuable resource for benchmarking future detection models. Releasing this dataset would have a broad impact on the community.

2.The F-SAT method is an important innovation, targeting high-frequency features that are critical for detection but vulnerable to adversarial manipulation. This frequency-focused adversarial training enhances model robustness without compromising accuracy on clean data, addressing a key gap in existing deepfake detection methods.

3.Comprehensive Experimental Evaluation:
   The experimental design is extensive, evaluating performance across standard benchmarks (ASVspoof2019 and WaveFake) as well as the authors' own test dataset. F-SAT demonstrates clear improvements in robustness across multiple corruption and adversarial attack scenarios. The addition of an ablation study further supports the effectiveness of the proposed method.

4. Extending RandAugment from image processing to audio is an inventive adaptation that helps improve model robustness on both clean and corrupted audio. This demonstrates the authors' resourcefulness in leveraging existing techniques and could be beneficial for future work in audio data augmentation.

**Weaknesses:**

1. The paper does not specify whether baseline models were subjected to adversarial training. If only the F-SAT model received this enhancement, it could bias the results. Including adversarially-trained versions of baseline models using contemporary adversarial methods would provide a fairer comparison and highlight F-SAT’s unique advantages.

2. While F-SAT’s focus on high-frequency components is intriguing, the rationale behind the reliance on high frequencies for detecting deepfake audio could be further elaborated.

3.Adversarial training, especially in the frequency domain with iterative updates, can be computationally demanding. Assessing F-SAT's efficiency, particularly compared to baseline models, would improve the paper's practicality.

**Questions:**

1.Given that F-SAT focuses on high-frequency perturbations, have the authors considered whether these perturbations might be perceptible to human listeners?




2. Were all baseline models subjected to similar adversarial training procedures as the proposed F-SAT model? Consistency in adversarial training across baseline models is essential to ensure a fair comparison of robustness improvements. If not, would the authors consider including adversarially trained baselines in future comparisons?

3 . How sensitive is F-SAT to the choice of hyperparameters, particularly the frequency ranges and perturbation magnitudes used for adversarial training?

---

> ### Author Response · Authors · 2024-11-25
> **Thank you for your thoughtful review and constructive suggestions**
>
> We thank the reviewers for their insightful feedback. We are glad that the reviewers recognize the broad impact that releasing the DeepFakeVox-HQ dataset would have on the community. Additionally, our robust training method has been highlighted as a significant advancement in deepfake detection. We address further details and reviewer suggestions in the sections below.
>
> **W1 - Baseline models using contemporary adversarial methods would provide a fairer comparison**
>
> Thank you for your question. RawNet3 is the best baseline model among all the baseline we study. As reviewer suggested, we've applied standard adversarial training to RawNet3 and compared it with F-SAT in Table 3. F-SAT exceed  standard Adversarial training method by 9% on orign data and by Average of 43% on attacked data.
>
> | Approach                      | Origin Real | Origin Fake | Origin Avg | Attack (Time) Real | Attack (Time) Fake | Attack (Time) Avg | Attack (Frequency) Real | Attack (Frequency) Fake | Attack (Frequency) Avg |
> |-------------------------------|-------------|-------------|------------|---------------------|---------------------|--------------------|--------------------------|--------------------------|-------------------------|
> | RawNet3+AT(Time)     | 94.7%       | 83.1%       | 88.9%      | 87.1%              | 12.4%              | 49.8%             | 66.5%                   | 16.5%                   | 41.5%                  |
> | RawNet3+F-SAT        | **97.5%**       | **98.4%**   | **97.9%**      | **90.2%**              | **87.0%**          | **88.6%**         | **93.3%**               | **92.8%**               | **93.1%**              |
>
> We will include experiments that apply adversarial training to all baseline models in the revised manuscript.
>
> **W2 - The rationale behind the reliance on high frequencies**
> The key rationale for our algorithm is that high frequency features help distinguish deepfake audio but are vulnerable to attacks, while low frequency features, though robust, are insufficient for training an effective detector. Our rationale is supported by three experiments:
>
> 1. We find the state-of-art Detection Model automatically rely on high frequencies for decision-making, as shown in Figure 2.
> 2.	Figure 8(b) （Figure 7b in revised version） demonstrates that adversarial attacks on high frequencies reduce model performance more significantly, suggesting their vulnerability. Additionally, human ears are less sensitive to high frequencies, making those attacks even harder to notice, highlighting the need for secure high-frequency features.
> 3. Low frequencies feature alone cannot adequately distinguish deepfake audio. As indicated in the table, employing a Biquad-lowpass filter to remove high frequency features during training and testing will reduces accuracy on original data.
>
> | Approach           | Origin Real | Origin Fake | Origin Avg |
> |--------------------|----------|----------|---------|
> | RawNet3+RandAug    | 97.6%    | 97.0%    | 97.3%   |
> | + Biquad-lowpass    | **98.9%**（+0.7%）| 86.5%（-10.5%）   | 92.7%（-4.6%）|
> | + F-SAT             | 97.5% （-0.1%）   | **98.4%** (+1.4%)  | **98.0%** (+0.7%) |
>
> Therefore, to develop a powerful and robust detector, retaining high-frequency features is essential, but they must be secured. This is the rationale behind our F-SAT approach.
>
> **W3 - F-SAT's training efficiency**
>
> Thank you for your advice. We have included an evaluation of F-SAT's training efficiency in the appendix. F-SAT’s training efficiency is influenced by hyperparameters such as attack iterations and restart counts, which identify the worst-case perturbation. Drawing on insights from "Fast Is Better Than Free: Revisiting Adversarial Training," we optimized these parameters by setting restarts to one and attack iterations to one or two, while employing a larger attack magnitude to enhance robustness.  Training time is shown in the table below.
>
> | Description                | w/o Adversarial Training | Standard Adversarial Training |  F-SAT |
> |----------------------------|---------------|--------------------------------|------------|
> | Training Duration (Days)   | 2             | 4.5                            | 8          |
> | Number of Epochs           | 15            | 15                             | 15         |
> | Hardware Used              | Single A100 GPU      |  Single A100 GPU                       | Single A100 GPU   |
>
>
> Although F-SAT requires longer training times, it improves accuracy by an average of 9% on original data and 43% on attacked data compared to Standard Adversarial Training. We should not compromise accuracy merely to accelerate training.

---

> ### Author Response · Authors · 2024-11-25
> **Continue**
>
> **Q1 - Whether attack is perceptible to human listener**
>
> Thank you for bringing this up. The attack we use to evaluate model robustness in our paper is imperceptible. We confirmed this through a human study where we presented 20 pairs of attacked and unattacked audio clips to 10 participants, who were asked to identify the attacked ones. The average prediction accuracy was 55% with a standard deviation of 7.58%, indicating that the attacked audio is nearly indistinguishable from the unattacked, resembling random guessing.
>
> Additionally, following Reviewer EGHT's advice, we utilized Signal-to-Noise Ratio (SNR) to quantitatively assess the perceptibility of the attack. An SNR greater than 40 dB indicates that the attack is almost imperceptible.
>
> |                       |  Mean  | Standard Deviation | Minimum | Maximum |
> |-----------------------|-------|---------------------|---------|---------|
> | Frequency Domain Attack | 68.76 | 5.80                | 40.07   | 84.83   |
> | Time Domain Attack      | 58.39 | 5.38                | 32.14   | 71.07   |
>
> **Q2 - Consistency in adversarial training across baseline models.**
>
> Thank you for your question. Among all the baseline models we evaluated, RawNet3 emerged as the best. In Table 3, we've applied standard adversarial training (AT) to RawNet3 and compared it with F-SAT. F-SAT significantly outperformed standard AT on both unattacked and attacked data.
>
> In the revised manuscript, we will include experiments applying adversarial training to all baseline models.
>
> **Q3 - How sensitive is F-SAT to the choice of hyperparameters, particularly the frequency ranges and perturbation magnitudes**
> Thank you for highlighting this. We have included an ablation study of hyperparameters, such as frequency ranges and magnitudes, in Figure 9 (Figure 8 in revised version) of the main paper. The results demonstrate that our approach is not sensitive to these parameters.

---

> > ### Comment · Reviewer_ijUY · 2024-11-26
> > **Thanks for the response.**
> >
> > Thank you very much to the authors for addressing the concerns I raised. The issues I was concerned about have been satisfactorily resolved. Considering the overall quality and contribution of the paper, I am willing to raise my score to 8.

---

### Official Review · Reviewer_EGHT · 2024-11-03

**Soundness:** 2
**Presentation:** 3
**Contribution:** 3
**Rating:** 8
**Confidence:** 4

**Summary:**

The paper attempts to improve deepfake detection by (1) proposing a large training and evaluation dataset called DeepFakeVox-HQ containing diverse synthetic and real speech recordings, (2) proposing a data augmentation method similar to randaugment for deepfake detectors and (3) proposing a frequency-selective adversarial training (F-SAT) method to make deepfake detectors more robust to adversarial attacks.

DeepFakeVox-HQ is a large dataset containing real and synthetic speech from existing datasets, speech generated by SOTA speech synthesis models as well as deepfakes found in-the-wild (on social media, etc.). Results show that models trained on DeepFakeVox-HQ generally perform better on existing deepfake while the models trained on the existing datasets have weak performance on DeepFakeVox-HQ, which indicates that DeepFakeVox-HQ includes information that prior works do not provide. DeepFakeVox-HQ will likely be a useful resource in deepfake research.

The proposed RandAugment scheme for deepfake detection utilizes a large bank of audio augmentations during training and yields significant improvements in deepfake detection accuracy.

The key contribution of F-SAT is to add adversarial perturbations to only certain frequency bands, which apparently results in lesser degradation of accuracy on un-perturbed data while providing greater robustness than standard adversarial training.

**Strengths:**

1. Paper is generally well-written and easy to read but some important details are missing
1. DeepFakeVox-HQ is a novel dataset containing data from prior datasets as well as novel deepfakes generated from SOTA speech synthesis models. I appreciate that the authors have curated a test set containing deepfake generation methods not covered in the training set _as well as deepfakes gathered from the internet_.  I encourage the authors to consider uploading the dataset to a platform like Huggingface Hub.
1. The proposed randaugment data augmentation method is effective at improving deepfake detection for RawNet3 models and is likely to be widely adopted if the source code is easy to use (I looked at the README in the attached supplemental material but did not find any instructions for the augmentation).
1. The proposed adversarial training method improves deepfake detection accuracy on clean and adversarially perturbed recordings (though I have some reservations regarding the experimental setup).

**Weaknesses:**

1. Some important details about the proposed approaches are not mentioned in the paper.

   1. The value of $\epsilon$ and $p$ (or $q$) used in adversarial training methods should be mentioned in the main body of the paper. Currently, it is mentioned in the caption of a table in the appendix
   1. The settings used for adversarial attacks during AT, F-SAT and evaluation need to be mentioned.
   1. The parameters of the augmentations used in randaugment need to be mentioned at least in the appendix.
   1. Detailed composition of DeepFakeVox-HQ needs to be mentioned including

      1. the method used for generating deepfakes (particularly noisy deepfakes),
      1. the number of audios from each deepfake generation system,
      1. the demographic distribution of the real and fake speakers,
      1. the number of utterances used from each of the datasets from prior works,
      1. quantitative measurement of synthetic speech quality using metrics like DNSMOS, or NORESQA.

1. The choice of accuracy as a metric seems to be inappropriate for a binary classification task. I would suggest using F1-score and equal error rate as the metrics. Moreover, reading tables and plots with two accuracy metrics for accuracy is a little confusing.
1. Conducting adversarial attacks in the frequency domain and reverting to the temporal domain is not novel and has been done before [2].
1. There is no comparison with other adversarial defenses for audio models. Many of the defenses created for speech and speaker recognition will also apply to the deepfake detection scenario. One method that is quite simple is [3]
1. The common practice is to use signal-to-noise ratio (SNR) as the bound for adversarial attacks in the audio domain [1] instead of $\ell_p$ bounds. I would highly recommend the authors use SNR as well. It is fairly straightforward to convert SNR to $\ell_2$ bounds and vice-versa. The main advantage of using SNR is that one has an idea how _perceptible_ the adversarial attack is.

1. Clarity issues:
   1. The caption Figure 9 needs to state that the results are of F-SAT
   1. Add results for 0-8K in Figure 8b


[1] Carlini, Nicholas, and David Wagner. "Audio adversarial examples: Targeted attacks on speech-to-text." 2018 IEEE security and privacy workshops (SPW). IEEE, 2018.

[2] Koerich, Karl Michel, et al. "Cross-representation transferability of adversarial attacks: From spectrograms to audio waveforms." 2020 International Joint Conference on Neural Networks (IJCNN). IEEE, 2020.

[3] Olivier, Raphael, Bhiksha Raj, and Muhammad Shah. "High-frequency adversarial defense for speech and audio." ICASSP 2021-2021 IEEE International Conference on Acoustics, Speech and Signal Processing (ICASSP). IEEE, 2021.

**Questions:**

1. What frequency range is the spec-magnitude attack being applied over in Figure 8a?
1. Do you have any _formal_ explanation for why the time-domain attack is less successful than the spec-magnitude attack? To me this seems counter-intuitive because the STFT is a linear and (mostly) invertible function so, from the perspective of the optimization, it should not matter if the attack was computed in the frequency domain or the time domain. I would be very interested in seeing more explanation for why the time-domain attack is unable to reach the solution acheived by the frequency-domain attack. Please also provide the detailed settings for all the adversarial attacks (time, frequency and phase domain) used in AT, F-SAT and during evaluation.
1.In principle, a frequency selective adversarial attack could be constructed entirely in the time domain by applying a band-pass filter to the adversarial perturbation after each optimization step (i.e. include the BP filter as part of the projection operation). This might be less computationally intensive than the proposed approach. Can you provide some discussion on why the proposed approach was favored?
1. Why is the performance of the model trained on DeepFakeVox-HQ so low on the In-the-wild dataset (see Figure 3)

---

> ### Author Response · Authors · 2024-11-25
> **Thank you for your thoughtful review and suggestions on the experiments**
>
> We thank the reviewer for their thoughtful feedback. We are glad that the reviewer finds our dataset to be significant and worthy publishing, as well as the effectiveness of our approach to improve deepfake dectection. We address the reviewer’s questions below:
>
> **W1.1 - The settings used for adversarial training during AT, F-SAT**
>
> Thank you for the suggestion.  We have included the training hyperparameters for AT and F-SAT in our revised version, as shown in the table below.
>
> | **Method** | **Type** | **Epsilon** | **Alpha** | **Attack Iter** | **Restart** | **Gamma (Loss Ratio)** |
> | ---------- | -------- | ----------- | --------- | --------------- | ----------- | ---------------------- |
> | AT         | l_inf    | 1.00E-4     | 4.00E-5   | 2               | 1           | 0.1                    |
> | F-SAT      | l_inf    | 5.00E-3     | 2.00E-3   | 2               | 1           | 0.1                    |
>
> **W1.2 - Attack Settings for Evaluation and SNR-based Perceptibility Assessment**
>
> Thank you for the suggestion. We have detailed the attack settings for evaluation in the table below and included them in our revised paper.
> | **Attack**       | **Type** | **Epsilon**               | **Alpha**                 | **Iter** | **Restart** | **SNR (mean)** | **SNR (std)** |
> |------------------|----------|---------------------------|---------------------------|----------|-------------|----------------|---------------|
> | Waveform(Time)     | l_inf | 1.00E-4 | 4.00E-5 | 2        | 1           | 58.4           | 5.4           |
> | Magnitude (Frequency)    | l_inf | 1.00E-3 | 4.00E-4 | 2        | 1           | 68.7           | 5.8           |
> | Phase (Frequency)        | l_inf | 2.00E-1                  | 1.00E-1                  | 2        | 1           | 39.7           | 4.9           |
>
> **W1.3 - Randaug Hyperparameter**
>
> Thank you for the advice. We have included the parameters of the augmentations used in RandAugment in the appendix. In our experiments, we configured aug_num=1 and set aug_prob=0.9.
>
> **W1.4.1 - Generate Deepfakes**
>
> Thank you for the advice. We have included the Dataset Details in appendix. We use the list of deepfake models (XTTS v2, StyleTTS v2, Metavoice, Whisperspeech, Vokan-TTS, and VoiceCraft and Elevenlabs) to generate deepfake voice for training set. to generate voices for the training set. Additionally, we employ Cosyvoice, PlayHT 2.0, Resemble, LOVO AI, and Lipsynthesis to create deepfake voices for the test set. we generate the noisy deepfakes with postprocessing augmentations.
>
> **W1.4.2 - The number of audios from each deepfake generation system**
>
> We utilize four real datasets—VCTK (12.0k), Librispeech-clean-100 (28.5k), Audioset (narration) (12.2k), and In-The-Wild (real parts: 9.3k)—to generate deepfake voices for the training set, with the number of samples from each source listed in the table below. For the test set, each source contributes 200 samples.
> | **Model**         | **metavoice** | **StyleTTS v2** | **XTTS v2** | **VoiceCraft** | **Whisperspeech** | **Vokan-TTS** | **Elevenlabs** |
> |-------------------|---------------|-----------------|-------------|----------------|-------------------|---------------|----------------|
> | **Samples**       | 61.7k           | 61.6k             | 61.8k         | 59.4k            | 61.9k               | 61.6k           | 3.2k           |
>
> **W1.4.3 - the demographic distribution of the real and fake speakers**
>
> We are sorry about that. Our dataset, drawn from multiple sources, makes it difficult to calculate the overall demographic distribution of speakers. However, for the deepfake audio we generated, the speaker distribution is a composite of VCTK, In-the-wilds, Audioset (narration), and Librispeech. This is because each source uses the same sentence inputs as the real audio.
>
> **W1.4.4 the number of utterances used from each of the datasets from prior works**
>
> The details of the reference datasets used are provided in the table below.
> | **Real**                       | **Fake**                   |
> |--------------------------------|----------------------------|
> | Libri-clean-100 28k            | Wavefake (English) 134k    |
> | Libri-clean-360 104k           | ASRspoof2019-train 2k      |
> | Libri-other-500 149k           | ASRspoof2019-dev 2k        |
> | Audioset (narration) 12k       | ASRspoof2019-eval 6k       |
> | VCTK 12k                       |                            |
> | In-The-Wild 9k                 |                            |
> | ASRspoof2019-train 2k          |                            |
> | ASRspoof2019-dev 2k            |                            |
> | ASRspoof2019-eval 7k           |                            |
> | VoxCeleb1 149k                 |                            |
> | **Total** 474k                 | **Total** 144k             |

---

> ### Author Response · Authors · 2024-11-25
> **Continue**
>
> **W1.4.5 - Quantitative measurement of synthetic speech quality using metrics like DNSMOS, or NORESQA.**
>
> Thanks for your advice, we have used DNSMOS [1] to quantitatively measure synthetic speech quality across these models, on a scale from 1 to 5, where higher values indicate better quality. We utilized sections of the VCTK and In-The-Wild datasets for this evaluation to assess model performance under both clean and noisy conditions.
>
> **Table: MOS Scores for VCTK Speaker_id: p244**
> **Table for VCTK Speaker: p244**
> |  **Model**      | **Ovrl MOS** | **Sig MOS** | **Bak MOS** | **P808 MOS**|
> |-----------------|----------|---------|---------|----------|
> | Real refer | 3.26     | 3.56    | 4.04    | 3.61     |
> | metavoice   | **3.29** | **3.58**| 4.05    | 3.63     |
> | StyleTTS v2 | 3.28     | 3.56    | **4.08**| **3.87** |
> | XTTS v2     | 3.13     | 3.41    | 4.00    | 3.78     |
> | VoiceCraft  | 3.16     | 3.51    | 3.94    | 3.61     |
> | Whisperspeech | 3.28   | 3.56    | 4.07    | 3.82     |
> | Vokan-TTS   | 3.23     | 3.55    | 4.01    | 3.71     |
>
>
> **Table for In-the-Wild Speaker: Alan Watts**
> | **Model**      | **Ovrl MOS** | **Sig MOS** | **Bak MOS** | **P808 MOS** |
> |----------------|--------------|-------------|-------------|--------------|
> | Real refer     | 3.02         | 3.40        | 3.74        | 3.57         |
> | metavoice      | 3.15         | 3.52        | 3.88        | 3.55         |
> | StyleTTS v2    | **3.28**     | **3.57**    | **4.06**    | **3.83**     |
> | XTTS v2        | 3.11         | 3.41        | 3.98        | 3.70         |
> | VoiceCraft     | 3.01         | 3.34        | 3.80        | 3.43         |
> | Whisperspeech  | 3.15         | 3.44        | 3.99        | 3.59         |
> | Vokan-TTS      | 2.94         | 3.39        | 3.66        | 3.60         |
>
> [1] Reddy, C. K., Gopal, V., & Cutler, R. (2021, June). DNSMOS: A non-intrusive perceptual objective speech quality metric to evaluate noise suppressors. In ICASSP 2021-2021 IEEE International Conference on Acoustics, Speech and Signal Processing (ICASSP) (pp. 6493-6497). IEEE.
>
> **W2 - The choice of accuracy as a metric**
> Thank you for the suggestion. We will include both F1-score and EER in our revised version. The accompanying table presents comparisons with other defenses, detailing F1 and EER results. These trends align with our original performance metrics and relate to our response in reply-W4 to you.
>
> | Model                  | Origin F1 | Origin EER | Attack (Time) F1 | Attack (Time) EER | Attack (Frequency) F1 | Attack (Frequency) EER |
> |------------------------|-----------|------------|---------------|----------------|---------------|----------------|
> | RawNet3+RandAug        | 0.9729    | 0.0238     | 0.6932        | 0.3048         | 0.6308        | 0.3671         |
> | +AT(Time)              | 0.8799    | 0.1175     | 0.2105        | 0.6921         | 0.2184        | 0.6829         |
> | + Gaussian smoothing   | 0.5267    | 0.4206     | 0.5424        | 0.4397         | 0.4903        | 0.5306         |
> | + MAD smoothing        | 0.9366    | 0.0603     | 0.5880        | 0.4079         | 0.5712        | 0.4298         |
> | + F-SAT                | 0.9810    | 0.0190     | 0.9048        | 0.0876         | 0.9297        | 0.0694         |
>
> **W3 - previous work relevant to adversarial attacks in the frequency domain and reverting to the temporal domain**
>
> Thank you for bringing up the paper “Cross-representation Transferability of Adversarial Attacks.” While that study focuses on adversarial attacks, our work concentrates on defense. Additionally, our research conducts an in-depth study on the impact of adversarial training across various frequency domains, providing more insights and achieving state-of-the-art robustness results. We have cited and discussed this paper in the related work section of our revised version.

---

> ### Author Response · Authors · 2024-11-25
> **Continue**
>
> **W4 - Compared with other defense method**
>
> Thank you for directing our attention to the paper titled "High-frequency Adversarial Defense for Speech and Audio." [1] Following your suggestion, we compare ours methods with [1] and the baselines used in [1], including Gaussian Smoothing [3], and adversarial training [2] in the time domain—techniques. We employed the same parameters as those in [1]. The results of these comparisons are outlined below. As the table shows, our method outperforms all other methods.
>
> | Approach               | Ori Real | Ori Fake | Ori Avg | Att(T) Real | Att(T) Fake | Att(T) Avg | Att(F) Real | Att(F) Fake | Att(F) Avg |
> |------------------------|----------|----------|---------|-------------|-------------|------------|-------------|-------------|------------|
> | RawNet3+RandAug        | **97.6%**    | 97.0%    | 97.3%   | 74.7%       | 66.0%       | 70.4%      | 63.0%       | 62.4%       | 62.7%      |
> | +AT(Time)              | 94.7%    | 83.1%    | 88.9%   | 87.1%       | 12.4%       | 49.8%      | 66.5%       | 16.5%       | 41.5%      |
> | +Gaussian smooth       | 57.8%    | 49.4%    | 53.6%   | 53.2%       | 51.3%       | 52.2%      | 46.2%       | 51.4%       | 48.8%      |
> | +MAD smoothing         | 96.2%    | 91.4%    | 93.8%   | 60.2%       | 59.0%       | 59.6%      | 56.3%       | 57.4%       | 56.8%      |
> | +F-SAT (Ours)          | 97.5%    | **98.4%**    | **98.0%**   | **90.2%**       | **87.0%**       | **88.6%**      | **93.3%**       | **92.8%**       | **93.1%**      |
>
> [1] Olivier, Raphael, Bhiksha Raj, and Muhammad Shah. "High-frequency adversarial defense for speech and audio." ICASSP 2021-2021 IEEE International Conference on Acoustics, Speech and Signal Processing (ICASSP). IEEE, 2021
>
> [2] Aleksander Madry, Aleksandar Makelov, Ludwig Schmidt, Dimitris Tsipras, and Adrian Vladu, “Towards deep learning models resistant to adversarial attacks,” international conference on learning representations, 2018.
>
> [3] Jeremy Cohen, Elan Rosenfeld, and Zico Kolter, “Certified adversarial robustness via randomized smoothing,” in Proceedings of the 36th International Conference on Machine Learning, Kamalika Chaudhuri and Ruslan Salakhutdinov, Eds. 09– 15 Jun 2019, vol. 97 of Proceedings of Machine Learning Research, pp. 1310–1320, PMLR.
>
> **W5 - SNR to evaluate how perceptible the adversarial attack is**
>
> Thank you for the suggestions. We have incorporated Signal-to-Noise Ratio (SNR) to assess the perceptibility of adversarial attacks in our main paper. The SNR for attacks in the time domain is approximately 58.4, while in the frequency domain it is 68.7. These values indicate the imperceptibility of the attacks.
>
>
> **W6 - Clarity issues**
>
> Thank you for the suggestions, we have revised the caption for Figure 9 (Figure 8 in revised version) and added results for the 0-8K range in Figure 8b (Figure 7b in revised version).
>
> **Q1- What frequency range is the spec-magnitude attack applied over in Figure 8a?**
>
> The spec-magnitude attack is applied across a frequency range of 0-8 kHz.
>
> **Q2 - Why Use F-SAT Instead of Adversarial Training in the Time Domain and Employ a Bandpass Filter for Post-Processing Perturbations?**
>
> Thank you for your question. Our F-SAT approach targets the magnitude component of audio signals, not affecting the phase. This focused strategy empirically enhances the effectiveness of our method.
>
> Adversarial training in the time domain has been shown to be ineffective, as evidenced by our experiments (referenced in Table 3) and supported by reference paper, "High-frequency adversarial defense for speech and audio" (Table 1). The ineffectiveness stems from time-domain attacks impacting both the magnitude and phase of the audio. Even using a bandpass filter does not alleviate the phase change. Our studies show that phase-focused adversarial training in the frequency domain degrades the performance of the RawNet3 model on natural data.
>
> **Q3 - Why is the performance of the model trained on DeepFakeVox-HQ so low on the In-the-wild dataset**
>
> Training on all datasets except ‘In-the-wild’ and testing on ‘In-the-wild’ leads to poor performance. Conversely, training on ‘In-the-wild’ and testing on all other datasets results in even worse outcomes. This is likely due to significant distributional differences in the ‘In-the-wild’ dataset. Factors such as the synthesis models used, the quality of reference audio, and other methods of generating deepfake samples contribute to these disparities.

---

> ### Author Response · Authors · 2024-11-25
> **Continue**
>
> **Other Concerns — Open Source of Data and Augmentation Instructions**
>
> We have outlined our data augmentation setup in the file dataset_aug_all.py, included in the supplemental materials. The hyperparameters for augmentation and detailed instructions for running the commands are provided in the README.
>
> Upon acceptance, we will open-source our code and data, and provide a comprehensive README to guide users.

---

> ### Comment · Reviewer_EGHT · 2024-11-25
> **Thank you for the comprehensive responses**
>
> Thank you for providing detailed responses. I find that my concerns have been address and I have increased my score.
>
> Please ensure that the tables below have been included and appropriately referenced in the manuscript, many of them have not yet been included. I have the following comments:
>
> W1.4.3 -- For several of these datasets the speaker demographic data is available. I would advise that that data be compiled into a table and included in the paper. My main concern is that if the data is heavily biased towards a specific demographic, then the generalizability of the results may be questionable. For example, if most of the data is from male speakers, does that impact the ability to detect deepfakes with female speakers?
>
> W4 -- please update results to use EER.
>
> I would recommend that all plots and tables in the main text use EER.
>
> Q2: Please include this explanation in the paper if it is not already there.

---

> > ### Author Response · Authors · 2024-11-28
> > **Thank you for your additional suggestions**
> >
> > Thanks for your advice.
> >
> > **W1.4.3**
> >
> > For demographic, we are using  the "wav2vec2-large-robust-24-ft-age-gender" model, available on Hugging Face, to analyze the age and gender distribution of our dataset. Due to the computational demands, this analysis requires several days to complete. We will include it in our appendix once finished.
> >
> >
> > **Q2**
> >
> > We have included explanation of F-SAT in revised paper (line 516-527).
> >
> >
> > **W4**
> >
> > We have updated all tables to display the F1 score in the main text, while both F1 and EER scores are provided in Appendix A.8. All tables and figures in main text are clearly readable right now.  We continue to report separate accuracy figures for real and fake samples, as it offers deeper insights. Specifically, 1. as datasets scale up, detecting fake audio proves more challenging than real audio because obtaining diverse real audio is always easier than generating diverse fake audio; 2. fake features concentrate at higher frequencies more than real features.
> >
> > Regarding the use of EER as a metric, we wish to highlight a potential limitation. Our experiments with RawNet3, trained using rand-augmentations (different noises, time stretch, pitch shift) and mixup on the ASVspoof2019 and tested on ‘In-the-Wild’ dataset [1], have shown results that greatly surpass current state-of-the-art methods. Although the EER is low, it does not reliably indicate detection effectiveness. When transferring from ASVspoof2019 to ‘In-the-Wild’, the distribution of real and fake samples shifted equally, causing the decision boundary to adjust from 0.5 to 0.9. This resulted in lower accuracy but an artificially improved EER. Therefore, we employ both F1 and EER to provide a more comprehensive analysis.
> >
> > | Approach                  | Real Acc | Fake Acc | EER       |
> > |---------------------------|----------|----------|---------|
> > | RawNet3, Noise, MixUp     | 0.01     | 0.99     | 0.11  |
> > | RawNet2 [1]                  | -        | -        | 0.34  |
> > | [2]                 | -        | -        | 0.24  |
> >
> > [1] Müller, Nicolas M., et al. "Does audio deepfake detection generalize?." arXiv preprint arXiv:2203.16263 (2022).
> >
> > [2] Yang, Yujie, et al. "A robust audio deepfake detection system via multi-view feature." ICASSP 2024-2024 IEEE International Conference on Acoustics, Speech and Signal Processing (ICASSP). IEEE, 2024.

---

### Author Response · Authors · 2024-11-25
**We ran the experiments asked by the reviewer and updated the paper. Edits are in blue.**

We sincerely thank all reviewers for their thoughtful and insightful feedback. We are encouraged by the recognition of the significance of our dataset and the effectiveness of our robust training methods. We ran all the experiments asked by the reviewers and integrated them to strengthen our paper. We indicate our paper edits in blue.

---

### Public Comment · ~Andrew_C._Cullen1 · 2024-12-05

Public comment re Ethics

As someone who is not reviewing this paper, I'd just like to flag up that the authors have been sourcing data from social media and YouTube, without appearing to acknowledge that this may be a) against the terms of service of these platforms and b) may constitute human involved research. Both of these factors may be considered as research practices that should require ethical oversight, and, at the very least, demonstration of having been issued appropriate licenses by copyright holders. These issues are exacerbated when one of the contributions of this paper is to produce a dataset that may be distributed, which is almost certainly against the terms of any licenses granted to the authors.

The Youtube terms of service explicitly state that users are not allowed to
- "access, reproduce, download, distribute, transmit, broadcast, display, sell, license, alter, modify or otherwise use any part of the Service or any Content except: (a) as expressly authorized by the Service; or (b) with prior written permission from YouTube and, if applicable, the respective rights holders;"
- "access the Service using any automated means (such as robots, botnets or scrapers) except (a) in the case of public search engines, in accordance with YouTube’s robots.txt file; or (b) with YouTube’s prior written permission;"
- "use the Service to view or listen to Content other than for personal, non-commercial use (for example, you may not publicly screen videos or stream music from the Service); or"

See: https://www.youtube.com/static?template=terms&gl=AU

All three of those above provisions could be seen to be violated by the authors scraping data from YouTube without a license.

Thanks

---

### Meta-Review · Area_Chair_KT4S · 2024-12-18

**Metareview:**

This paper makes significant advancements in deepfake detection through three key contributions: (1) Introduction of DeepFakeVox-HQ: A comprehensive dataset comprising high-quality synthetic and real speech recordings, designed to benefit the research community. (2) Development of an enhanced data augmentation method: Tailored to improve the performance of deepfake detectors. (3) Proposal of frequency-selective adversarial training to enhance detector robustness against adversarial attacks. The reviewers have recognized the paper as well-written, with the dataset being a valuable resource for the community and the proposed methods effectively improving detection accuracy. However, the final version should address the reviewers’ suggestions to enhance clarity and reproducibility. Key revisions include: (1) Adversarial training details: Adding the setup specifics for adversarial training and incorporating the Equal Error Rate (EER) as an evaluation metric. (2) Claims revision: Refining the claims of the paper’s contributions, as indicated by Reviewer erKc.
(3) Additional details: Providing more information in response to Reviewer NjHP's feedback. Overall, this work is meaningful to the community but requires these revisions to further ensure clarity and reproducibility.

**Additional Comments On Reviewer Discussion:**

All reviewers provided meaningful comments, checked the responses of the authors, and provided feedback to the authors.  At the first round of reviewing, reviewers agree the work is meaningful to the community, the problem is important, and the proposed methods are effective. The reviewers concerned about the missing setup details of adversarial training and incorrect claims. The authors' rebuttal addresses the concerns of reviewers well.

---

### Decision · Program_Chairs · 2025-01-22

Accept (Poster)